# Altered chromatin landscape and enhancer engagement underlie transcriptional dysregulation in MED12 mutant uterine leiomyomas

Mthabisi B. Moyo [1], J. Brandon Parker [2] & Debabrata Chakravarti [2,3,4 ✉]

Uterine leiomyomas (fibroids) are a major source of gynecologic morbidity in reproductive age women and are characterized by the excessive deposition of a disorganized extracellular matrix, resulting in rigid benign tumors. Although down regulation of the transcription factor AP-1 is highly prevalent in leiomyomas, the functional consequence of AP-1 loss on gene transcription in uterine fibroids remains poorly understood. Using high-resolution ChIP-sequencing, promoter capture Hi-C, and RNA-sequencing of matched normal and leiomyoma tissues, here we show that modified enhancer architecture is a major driver of transcriptional dysregulation in *MED12* mutant uterine leiomyomas. Furthermore, modifications in enhancer architecture are driven by the depletion of AP-1 occupancy on chromatin. Silencing of AP-1 subunits in primary myometrium cells leads to transcriptional dysregulation of extracellular matrix associated genes and partly recapitulates transcriptional and epigenetic changes observed in leiomyomas. These findings establish AP-1 driven aberrant enhancer regulation as an important mechanism of leiomyoma disease pathogenesis.

---

[1] Driskill Graduate Program in Life Sciences, Feinberg School of Medicine, Northwestern University, Chicago, IL 60611, USA. [2] Department of Obstetrics and Gynecology, Division of Reproductive Science in Medicine, Feinberg School of Medicine, Northwestern University, Chicago, IL 60611, USA. [3] Department of Pharmacology, Feinberg School of Medicine, Northwestern University, Chicago, IL 60611, USA. [4] Robert H. Lurie Comprehensive Cancer Center, Northwestern University, Chicago, IL 60611, USA. ✉email: debu@northwestern.edu

Uterine fibroids affect over 70% of all reproductive age women and are a major cause of gynecologic and reproductive dysfunction, ranging from profuse menstrual bleeding and pelvic discomfort to infertility, recurrent miscarriage, and preterm labor[1]. Leiomyomas are clonal tumors that originate from the smooth muscle layer of the uterine wall, the myometrium[2]. They are primarily characterized by an increased deposition of a disorganized extracellular matrix (ECM), resulting in benign neoplasms of varying size[3,4]. Despite a high prevalence, viable long-term non-surgical treatment options for uterine fibroids do not exist. Many women diagnosed with the disease ultimately undergo hysterectomies as the only curative treatment, with fibroid-related hysterectomies accounting for over 30% of all hysterectomies[5].

High-throughput sequencing techniques have provided major insights into the mechanisms of disease pathogenesis[6]. In uterine leiomyomas, whole-exome and whole-genome sequencing have identified multiple genetic aberrations that may be required for leiomyoma development and growth. Fibroid tumors stratify into four main subtypes that are dependent on the mutational status of mediator of transcription subunit 12 (*MED12*), fumarate hydratase (*FH*), high mobility group AT-hook 2 (*HMGA2*) translocations, and collagen (*COL4A5-COL4A6*) gene deletions[7]. In particular, the characterization of mutations in *MED12*, a key subunit of the mediator of RNA polymerase II transcription (Mediator) complex that globally regulates RNA polymerase II-dependent transcription, is of considerable interest as they have been found in ~70% of patients diagnosed with uterine leiomyomas, with mutations occurring predominantly in exon 2 of *MED12*[8]. In addition to whole-exome sequencing, gene expression profiling of leiomyomas has been performed, primarily using RNA microarrays[7,9,10]. Recent work has revealed subtype-specific gene expression profiles, indicating possibly different mechanisms of pathogenesis in fibroid tumors belonging to different subtypes[9]. Despite multiple gene expression studies identifying aberrant gene regulation, very little is known regarding the mechanisms of gene dysregulation in uterine leiomyomas.

Aberrant epigenetic modifications and transcription factor chromatin binding have long been implicated in transcriptional dysregulation and disease development[11–13]. In addition, recent studies demonstrate that changes in chromatin structure at enhancer regions also play a major role in disease pathogenesis[14]. However, the role of enhancer chromatin structure in uterine leiomyomas remains unexplored.

Enhancers are long-range *cis*-regulatory DNA elements (CREs) that spatially organize in close proximity to their respective promoters, thereby enabling direct contact between *trans*-acting factors and gene promoters[15,16]. Alterations in histone modification state as well as transcription factor and co-factor DNA binding affinity lead to changes in enhancer 3D architecture, resulting in alterations to RNAPII-dependent gene transcription. Recent chromosome conformation capture techniques have led to high-resolution maps of enhancer-promoter contacts, allowing for the unambiguous pairing of enhancers with their corresponding promoters[17,18]. In addition, promoter capture Hi-C (CHi-C) has been used to demonstrate the dynamism of enhancer-promoter contacts during cell differentiation[19,20]. However, the existence of dynamic or altered contacts in human disease as a result of epigenetic changes to chromatin has not been investigated.

Previous studies suggest that culturing of primary cells from tissue samples may partially alter the gene expression profiles of cells[21]. Additionally, in primary cell culture of cells obtained from leiomyoma patient tissue samples, a rapid loss of *MED12* mutant cells was observed, suggesting a very limited viability of this cell population in culture[22,23].

In this study, freshly procured tissue samples from women who have undergone hysterectomies as a course of treatment for uterine leiomyomas, confirmed to have a glycine-to-aspartate (G44D) or glycine-to-serine (G44S) substitution in exon 2 of MED12, are used to characterize epigenetic changes in the disease. Adjacent, non-diseased areas of the myometrium from the same patients are also collected to represent normal (wild-type [WT]) samples. In an effort to avoid artifactual alterations to the transcriptomic and epigenomic profiles of patient samples, gene expression profiling by RNA-sequencing (RNA-seq) as well as epigenetic profiling by high-resolution chromatin immunoprecipitation-sequencing (ChIP-seq) and promoter capture Hi-C are performed directly from tissue samples with minimal processing. Our integrative analysis of transcriptomic and epigenetic changes, highlighted by the near-native characterization of long-range promoter interactions in uterine fibroids, identifies differential transcription factor occupancy, differential enhancer engagement, and altered enhancer-promoter contacts as key events that drive gene dysregulation in leiomyomas.

## Results

**Transcriptome profiling of fibroids.** We used RNA isolation followed by massively parallel sequencing (RNA-seq) to examine the transcriptome profiles of normal myometrium (WT) and matched leiomyoma (G44D/S) tissue obtained from 15 women. A high degree of similarity between biological replicates of myometrium transcriptome profiles was seen, with a similar observation among biological replicates of leiomyoma tissue samples. Hierarchical clustering of all RNA-seq datasets highlights clustering primarily by disease state (Fig. 1a). Significantly, principal component analysis of the most variable genes revealed that 43% of the variance (PC1) is explained by the disease state, with biological replicates co-segregating based on tissue type (Fig. 1b). This suggests that the changes in gene expression between normal and *MED12* mutant disease tissue types are primarily attributable to biological pathways that are important for the development and maintenance of the leiomyoma disease state.

Differential gene expression analysis identified 5831 differentially regulated genes at a false discovery rate of 0.01 (Fig. 1c, Benjamini–Hochberg (BH) correction of Wald test determined $P$ values). Two thousand nine hundred and sixty-five genes exhibited a >2-fold change in expression between myometrium and leiomyoma samples, while 1014 genes were differentially expressed at >4-fold (Fig. 1d). Notably, dysregulated genes in leiomyoma samples are associated with ECM and collagen formation, organization, and degradation, which are key up-regulated biological processes in uterine fibroid disease pathogenesis (Fig. 1e, f and Supplementary Fig. 1a–e)[2].

While regulation of RNA levels occurs primarily at transcription, which includes epigenetic control of the RNAPII-mediated transcription machinery, RNA levels are also regulated post-transcriptionally through the regulation of RNA processing and degradation[24]. It has been demonstrated that differences in intronic read counts from RNA-seq experiments can serve as a measure of changes in transcriptional activity and even predict post-transcriptionally regulated genes[25]. To investigate possible post-transcriptionally driven processes in uterine leiomyomas, exon–intron split analysis (EISA) was carried out. Exonic and intronic read count changes between myometrium and leiomyoma tissue samples were highly correlated, indicating that gene expression changes seen in the disease state were primarily transcription driven and not post-transcriptional (Supplementary Fig. 1f, Pearson's $R = 0.89$). However, 331 genes were identified as possible targets of post-transcriptional regulation. Scanning of the 3′-untranslated regions (UTRs) of genes identified seed sequences

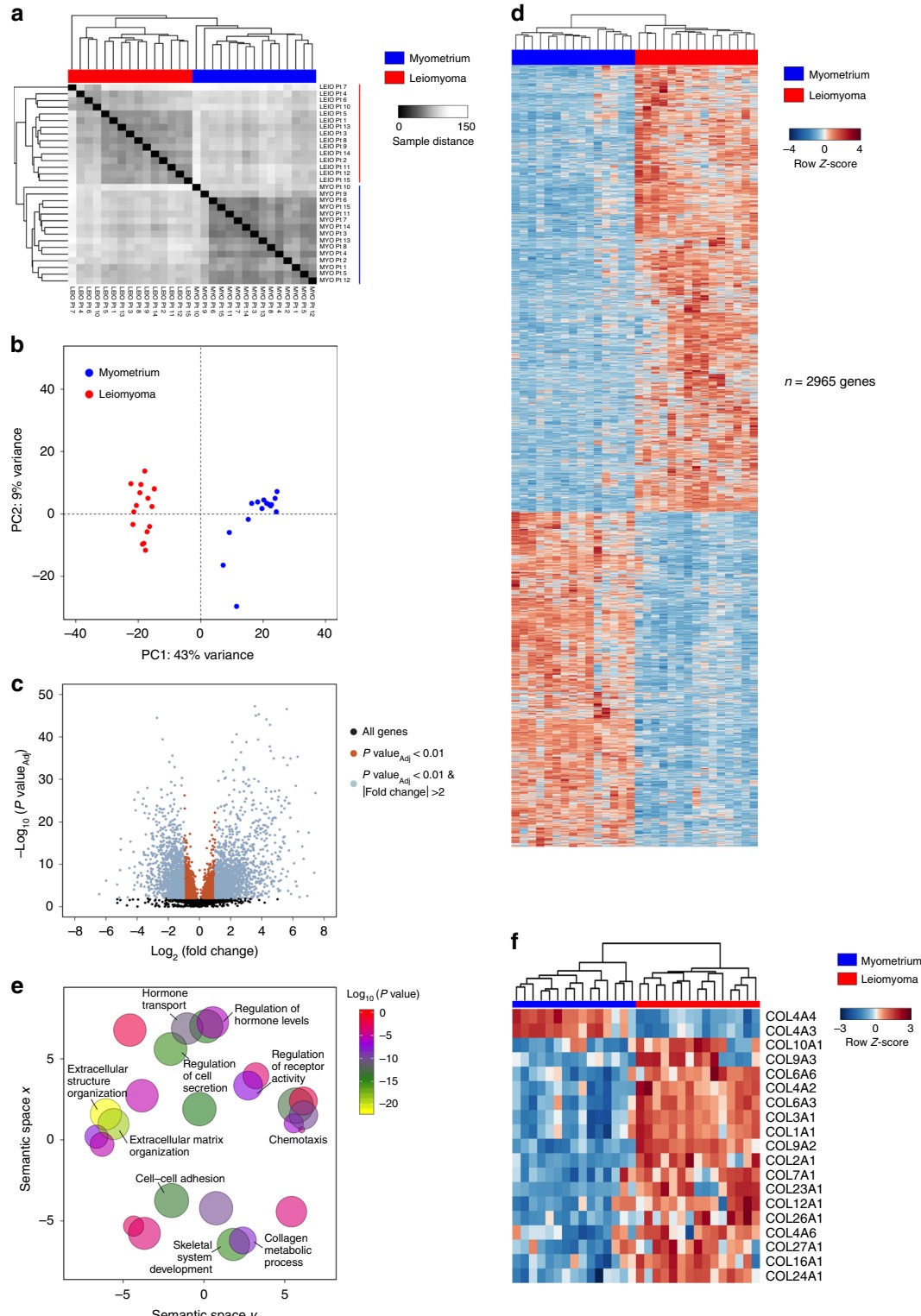

**Fig. 1 Transcriptome profiling reveals transcriptional dysregulation of key biological processes in fibroids. a** Heat map representing tissue sample Euclidean distance matrix clustering of RNA-sequencing profiles for 15 patients. **b** Principal component analysis plot of the top 500 most variable genes in myometrium and leiomyoma tissue samples. First two principal components are plotted. **c** Volcano plot of all differentially expressed genes (sienna, $n = 5831$) and differentially expressed genes with >2-fold change (gray, $n = 2965$) in myometrium and leiomyoma (FDR < 0.01, BH-corrected Wald test). Unchanged transcripts (FDR > 0.01) are shown in black. $P$ values are truncated at $1 \times 10^{-50}$ for visualization purposes. **d** Heat map of differentially expressed genes with >2 fold change in myometrium (blue) vs. leiomyoma (red) tissue samples. Gene expression levels relative to the mean expression are shown as row $Z$-scores ($n = 2965$ genes). **e** Scatter plot of confidence scores for enriched gene ontologies associated with differentially expressed genes, with ontologies clustered by functional similarity in the semantic space. **f** Hierarchically clustered heat map of differentially expressed collagen genes. Gene expression levels relative to the mean expression are shown as row $Z$-scores.

for mir-216 and mir-124a, which have both been demonstrated to play a role in tumor growth factor-β signaling, a major pathway implicated in tissue fibrosis (false discovery rate [FDR] < 0.05, BH-corrected Wald test)[2,26–28]. In summary, these results highlight perturbations, primarily at the transcriptional level, of a large proportion of the uterine muscle tissue transcriptome, the majority of which are dysregulated protein-coding genes, including ECM-associated genes.

**Disproportionate histone acetylation changes at intergenic regions.** Acetylation of the histone H3 tail at lysine 27 (H3K27) is a post-translational modification that is highly correlated with active transcription at enhancers and promoters[29]. To provide epigenetic mechanistic insight into the predominantly transcriptional changes in gene expression observed by leiomyoma transcriptome profiling, chromatin immunoprecipitation with massively parallel sequencing (ChIP-seq) of H3K27Ac was performed (Supplementary Table 1). Similarly to RNA-seq samples, biological replicates of H3K27 acetylation ChIP experiments are highly correlated and also cluster by tissue type (Fig. 2a). Significantly, 30% ($n = 16,752/54,488$) of all identified H3K27Ac regions are differentially acetylated with >2-fold change in signal and ~10% of sites ($n = 5840/54,488$) exhibit >4-fold change in H3K27Ac signal (Fig. 2b and Supplementary Fig. 2a).

Analysis of genomic loci at sites of altered H3K27Ac signal revealed changes at predominantly promoter-distal regions, as evidenced by a significant deviation from the expected H3K27Ac signal distribution across the genome (Fig. 2c and Supplementary Fig. 2b, ($\chi^2(4) = 1205$, $p < 0.05$). While gene promoter acetylation signal accounts for ~17% of all H3K27-acetylated regions in myometrium and leiomyoma, acetylated promoters account for only 8% of all differentially acetylated regions. Conversely, higher than expected numbers of promoter-distal sites were differentially acetylated, with 85% of all differentially acetylated regions occurring at intergenic and intronic regions, compared to 75% of sites in myometrium and leiomyoma. This suggests alterations at *cis*-regulatory elements such as enhancers in uterine leiomyomas.

Strikingly, while analysis of dysregulated gene promoters shows that H3K27Ac signal changes are positively correlated with differential gene expression, 46% of all dysregulated gene promoters show little to no change in promoter H3K27 acetylation in uterine leiomyomas (Fig. 2d). Identified examples of genes that, despite being overexpressed in leiomyomas, do not show a significant change in promoter H3K27 acetylation include fibronectin 1 (*FN1*), a disintegrin and metalloprotease domain-containing protein 19 (*ADAM19*), and collagen gene *COL12A1*, key genes involved in ECM formation and degradation (Fig. 2e, f and Supplementary Fig. 2c). In cases such as these, changes in enhancer activity, in addition to promoter acetylation changes, may together better explain the observed gene expression changes. Given the large proportion of the H3K27Ac cistrome that is modified in leiomyomas and the significant number of promoters of differentially expressed genes with little to no change in H3K27 acetylation, we predict that modifications in enhancer regions may play a role in uterine leiomyoma gene dysregulation.

**Enhancer malfunction leads to gene expression changes.** H3K27Ac ChIP-seq data suggest enhancer dysfunction as a likely defining feature of leiomyoma transcriptional dysregulation. Assignment of enhancers to target gene promoters using nearest-neighbor approaches has been shown to suffer from low accuracy[18,30]. Therefore, to assign dysregulated enhancers to target genes and determine if they are directly linked to differential gene expression in uterine fibroids, promoter capture

Hi-C was performed. In total, 191,104 high confidence promoter contacts were identified in myometrium and leiomyoma tissue samples, with contacts mapping to 98,957 unique promoter-distal regions (Supplementary Table 2, Supplementary Fig. 3a). High confidence promoter contacts significantly overlapped with H3K27-acetylated regions, including differentially acetylated H3K27 enhancer regions (Fig. 3a and Supplementary Fig. 3a, b).

Using promoter capture Hi-C to unambiguously assign enhancers to their corresponding promoters, we identified 2715 enhancers highly enriched or depleted of H3K27Ac signal that were directly linked with differentially expressed genes (Fig. 3b and Supplementary Fig. 3c). One thousand eight hundred and thirty-five enhancers were associated with induced or repressed genes exhibiting little to no change in promoter H3K27 acetylation. In addition, the magnitude in changes in enhancer acetylation signal was significantly greater than acetylation changes at dysregulated gene promoters. These results suggest that in addition to changes at promoters, altered H3K27Ac at enhancers plays a significant role in aberrant transcription of genes in uterine leiomyomas, including genes with unchanged promoter-proximal H3K27Ac signal.

Capture Hi-C studies have previously demonstrated a signal-dependent alteration of promoter contacts during differentiation[19,20]. We investigated whether promoter contact strength is also altered in a disease dependent manner in *MED12* mutant uterine fibroids. Eight thousand and seventy eight altered promoter contacts were identified, with 1974 contacts displaying >2-fold change in signal (Supplementary Table 2). Twenty five percent ($n = 2040/8078$) of contacts were associated with differentially expressed genes, which ontology analysis revealed to be enriched for genes involved in ECM organization (Fig. 3c, d). A survey of all altered contacts overlapping enhancers with differential H3K27Ac signal shows that regions with increased acetylation are associated with increased promoter contact strength, whereas regions depleted of H3K27Ac signal are linked to both contacts with increasing and decreasing strength (Fig. 3e). Approximately 42% (521/1229) of all altered contacts overlapping enhancers with differential H3K27Ac signal are associated with differentially expressed genes. Examples of key ECM-associated genes that demonstrate a change in gene expression accompanied by a change in promoter contact strength include *FN1*, *ADAM19*, ETS proto-oncogene 2 (*ETS2*), as well as collagen genes *COL6A3*, and *COL12A1* (Fig. 3f, g, Supplementary Figs. 3e, f and 4a).

Depending on the status of H3K27 acetylation and promoter contact strength, modified promoter-distal CREs can be classified into three distinct categories: (i) regions with unaltered promoter contact strength and differential H3K27 acetylation; (ii) regions with altered promoter contact strength and unchanged H3K27 acetylation; and (iii) regions with altered promoter contact strength and differential H3K27 acetylation (Supplementary Fig. 3d). Previous studies demonstrate that promoter contacts are cell-type-defining features, and as such, the majority of contacts are stable and unchanged[18,19,31]. However, despite the majority of contacts being stable, a subset are dynamic and do correlate with major events in the cell, such as changes in chromatin occupancy of transcription factors that are important in cell differentiation. Consistently, differentially acetylated enhancer regions with unaltered contact strength constitute the largest class of modified enhancers in leiomyomas, highlighting that enhancer architecture changes primarily consist of changes in histone modification signals. However, 27% of altered enhancer regions do show changes in promoter contact strength in the disease state. Although the function of altered contacts in uterine leiomyomas is unknown, the significance of altered contacts as demonstrated in cell differentiation may indicate a role for altered contacts in leiomyoma disease pathogenesis.

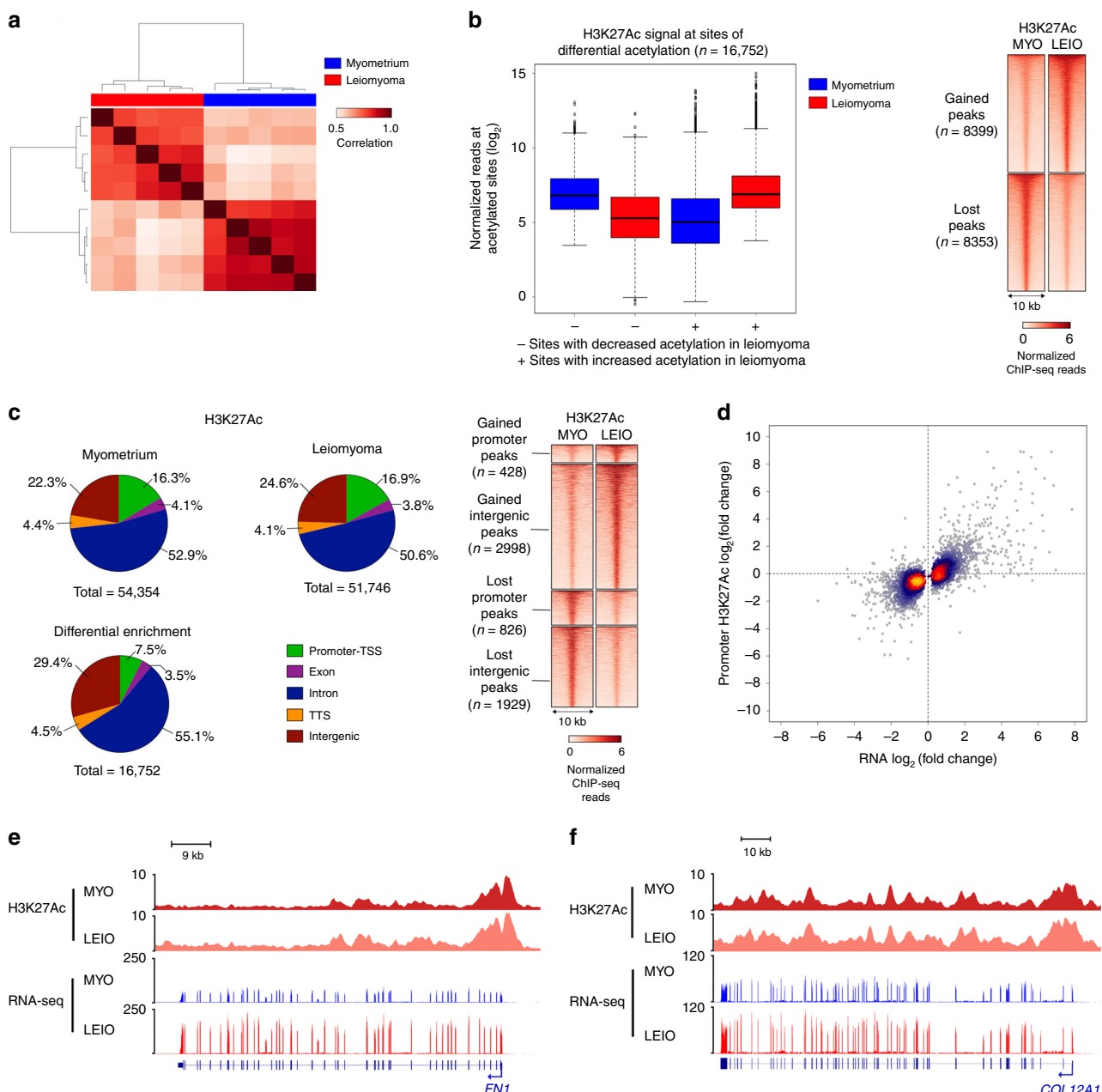

**Fig. 2 Histone acetylation changes occur disproportionately at intergenic regions. a** Correlation (Pearson's) heat map of H3K27Ac ChIP sample affinity scores obtained from myometrium (blue) and leiomyoma (red) ChIP-seq read counts. **b** Box and whisker plot of normalized reads at differentially acetylated sites in myometrium and leiomyoma (left panel). Plots of sites with decreased H3K27Ac signal (−) and increased signal (+) are shown separately. Centerline represents the median, bounds of box represent the interquartile range (IQR) and whiskers are 1.5 × IQR. Right panel shows a heat map of normalized H3K27Ac ChIP-seq reads at differentially acetylated regions in myometrium vs. leiomyoma. Heat map represents average signal of five biological replicates. **c** Pie charts of H3K27Ac signal distribution at genomic loci in myometrium (top left of left panel), leiomyoma (top right of left panel) and at differentially acetylated regions (bottom left of left panel). Right panel represents heat map of normalized H3K27Ac ChIP-seq reads at gene promoters and intergenic loci with differential H3K27Ac signal. Heat map represents average signal of five biological replicates. **d** Heat scatter plot of fold change in H3K27Ac signal against fold change in gene expression at promoters of differentially expressed genes ($n = 3559$). **e, f** Unchanged H3K27Ac signal in myometrium (red) and leiomyoma (salmon) at *FN1* (**e**) and *COL12A1* (**f**) genes. Normalized RNA reads for myometrium (blue) and leiomyoma (red) are also shown.

Capture Hi-C also revealed instances where decreased H3K27Ac signal in leiomyoma samples at an enhancer associated with a gene occurred concomitantly with an increase in H3K27Ac signal at another enhancer associated with the same gene. Six hundred and sixty-two genes were shown to exhibit such cases of differential enhancer usage, involving a total of 3884 enhancers. Differential enhancer usage was seen in both up- and down-regulated genes and occurred primarily at enhancers with stable,

unaltered promoter contacts, although a few cases were also observed in a subset of genes that have enhancers with altered contact strength (Supplementary Fig. 4c, d).

Promoter capture Hi-C performed on myometrium and leiomyoma tissue samples demonstrates multiple changes in enhancer architecture in uterine leiomyomas. This includes aberrant enhancer acetylation, which is correlated with changing gene expression for genes where promoter acetylation is

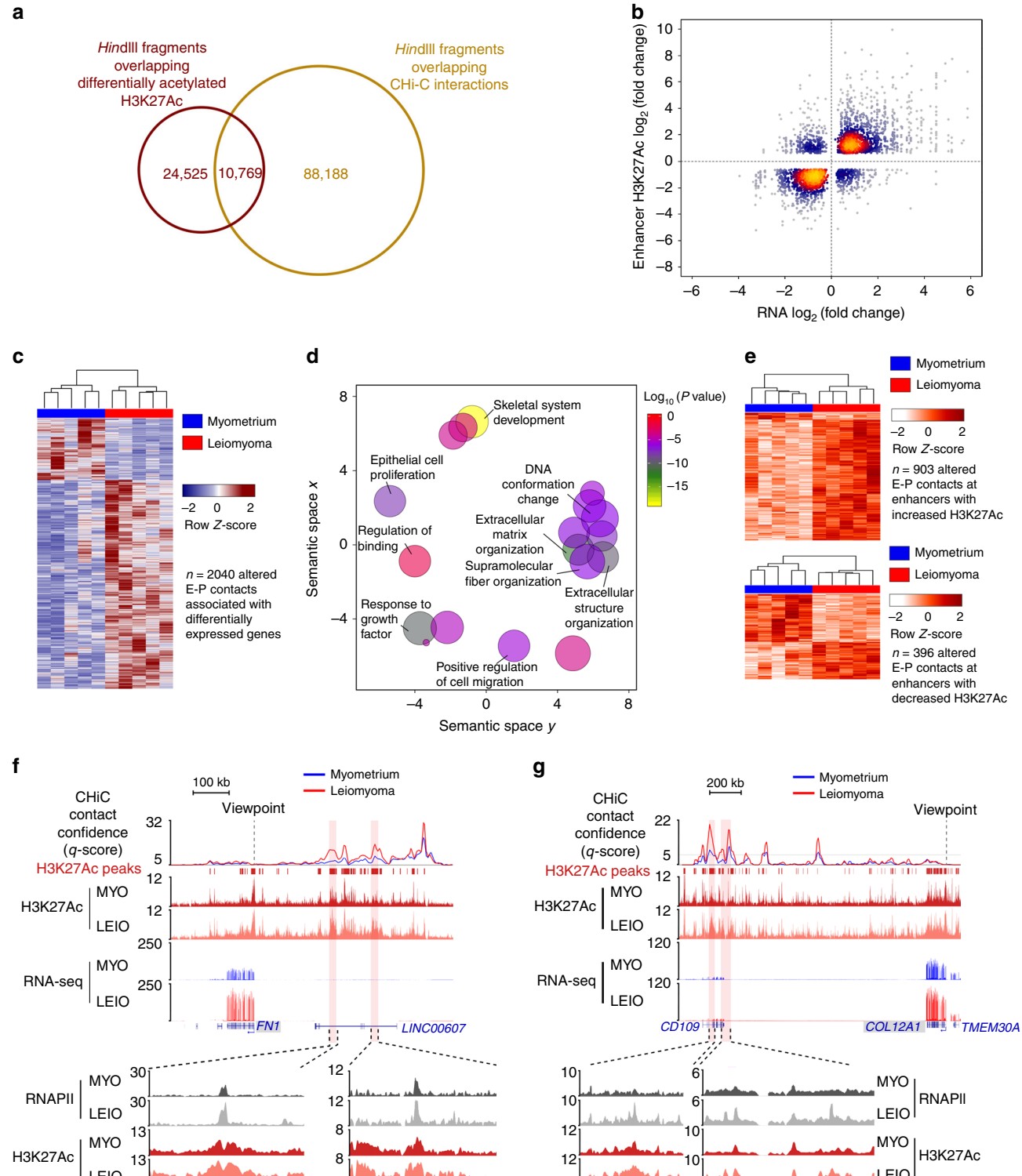

**Fig. 3 Enhancer malfunction leads to gene expression changes in leiomyomas. a** Venn diagram of promoter contacts overlapping differentially acetylated H3K27 regions in myometrium and leiomyoma tissue. *Hin*dIII digest fragments of the human genome are used to define unique, non-overlapping regions in the genome, to which H3K27Ac peaks and promoter contacts are then assigned. See CHi-C data processing and analysis in Methods. **b** Heat scatter plot of fold change in gene expression against fold change in H3K27Ac signal at enhancers associated with differentially expressed genes ($n = 2715$ enhancer regions). **c** Heat map of altered promoter contacts associated with differentially expressed genes ($n = 2040$ contacts). **d** Scatter plot of confidence scores for enriched gene ontologies associated with altered promoter contacts linked to differentially expressed genes, with ontologies clustered by functional similarity in the semantic space. **e** Heat maps of altered enhancer-promoter contacts that overlap with increasing H3K27Ac (top panel, $n = 903$ contacts) and decreasing H3K27Ac (bottom panel, $n = 396$ contacts) at enhancer regions in myometrium and leiomyoma tissues. **f, g** Genomic loci for *FN1* (**f**) and *COL12A1* (**g**) genes, which both have unchanged H3K27Ac signal at promoters. Altered confidence scores (CHiCAGO *q*-scores) are shown for myometrium and leiomyoma. In addition, H3K27Ac ChIP-seq and RNA-seq genomic tracks are shown, with H3K27Ac and RNAPII ChIP-seq signal in regions containing altered contacts highlighted (bottom zoomed in inset).

unchanged, as well as for genes with differential promoter acetylation. This highlights enhancer malfunction as an important mechanism of leiomyoma gene dysregulation, and in the case of differentially expressed genes with stable promoter H3K27Ac signal, provides a mechanism of gene dysregulation in the absence of promoter-proximal chromatin changes. The observation of altered contacts also suggests a partial rewiring of chromatin architecture in uterine leiomyomas.

**Alteration of AP-1 chromatin occupancy in leiomyomas**. Given the predominantly promoter-distal changes in H3K27 acetylation and the alterations in promoter contact strength, differentially acetylated enhancer regions were analyzed for the presence of transcription factor DNA-binding motifs that may be important in regulating gene transcription in uterine fibroids. The activator protein-1 (AP-1) transcription factor motif was enriched at differentially acetylated enhancers (Fig. 4a). Analysis of transcription factor motifs at promoters of differentially expressed genes also showed an enrichment of the AP-1-binding motif, suggesting AP-1-driven dysregulation of direct AP-1 target genes (Fig. 4b). Notably, expression of the *JUN*, *FOS*, and *ATF* families of genes was down-regulated in leiomyoma tissue (Fig. 4c and Supplementary Fig. 5a). This is consistent with previous studies highlighting greater than five-fold reduction in *FOS* and *JUN* mRNA levels in uterine fibroids[10,32]. Interestingly, AP-1 subunit genes exhibit little to no change in promoter H3K27Ac. A survey of enhancers associated with *JUN* and *FOS* showed differential H3K27Ac signal at a small subset of enhancers, with differential enhancer usage observed at enhancers contacting the *JUN* promoter, although promoter contact strength was unaltered at both *JUN* and *FOS* enhancers (Supplementary Fig. 5b, c). While the mechanisms of AP-1 gene regulation may be varied, these results suggest that down-regulation of AP-1 complex gene expression may be driven in part by changing enhancer chromatin architecture.

AP-1 motif enrichment at differentially acetylated promoter-distal regions, along with a loss of AP-1 subunit gene expression, suggests that AP-1 binding at enhancers could be perturbed in uterine leiomyomas. To test this, ChIP-seq of AP-1 subunits JUN and FOS was performed (Supplementary Table 1 and Supplementary Fig. 5d, e). Motif analysis of promoter-distal JUN- and FOS-bound sites confirmed that the majority of identified peaks overlap with AP-1 motifs (Supplementary Fig. 5f). Two thousand nine hundred and ninety-three out of 25,736 FOS peaks were differentially bound in leiomyoma tissue samples with 86% of altered binding sites ($n = 2573/2993$) showing a loss in binding affinity (Fig. 4d, Supplementary Fig. 6a). JUN binding was perturbed in a similar manner, with 2919 out of 18,283 JUN sites differentially bound and 95% of sites ($n = 2773/2993$) also showing a decrease in binding affinity (Fig. 4e and Supplementary Fig. 6b). A positive correlation between H3K27Ac signal changes and differential FOS/JUN occupancy at promoter-distal sites is also observed (Fig. 4f). Importantly, 761 enhancer regions with differentially bound FOS or JUN were identified as directly interacting with 420 differentially expressed gene promoters, which include ECM-associated genes and a subset of previously identified AP-1 target genes with promoter-proximal AP-1 motifs (Fig. 4g, h). A small subset of AP-1-depleted enhancer sites that coincided with altered promoter contacts were also identified, with an increase in JUN and FOS binding affinity being associated with an increase in promoter contact strength (Supplementary Fig. 6c, d). In contrast, a decrease in binding affinity was associated with both a gain and a loss in promoter contact strength. Examples of differentially expressed genes that show changes in enhancer binding of JUN and FOS include *ADAM19* and EGF-containing fibulin-like ECM protein 1 (*EFEMP1*) (Fig. 4i, j).

ChIP-seq of AP-1 subunits JUN and FOS in myometrium and leiomyoma tissue samples revealed a depletion of AP-1 on chromatin in leiomyomas, including depletion at a subset of enhancers that make contacts with promoters of dysregulated genes. From these results we conclude that AP-1 occupancy at H3K27-acetylated enhancers in myometrium is important in regulating uterine muscle cell gene regulatory programs and a loss of AP-1 binding at enhancers as a result of decreased AP-1 gene expression may lead to widespread gene dysregulation in leiomyomas.

**CDK8 subcomplex chromatin occupancy correlates with enhancer acetylation changes**. MED12 is a member of the four-subunit CDK8 submodule comprised of CDK8, MED12, MED13, and cyclin C. The submodule associates with core Mediator to form a stable CDK8-Mediator complex, which alters the transcription regulation function of Mediator[33,34]. Previously published interaction studies in heterologous cell lines expressing mutant MED12 (G44D/S) have suggested that mutations in exon 2 of MED12 may result in a decrease in cyclin C-CDK8 chromatin occupancy at sites of active transcription[35]. To assess whether exon 2 mutations at glycine 44 of MED12 result in a loss of CDK8 chromatin binding in uterine leiomyomas, ChIP-seq of CDK8 and MED12 was performed (Supplementary Table 1 and Supplementary Fig. 7a, c). Thirty thousand six hundred and ninety-seven genomic sites co-bound by CDK8 and MED12 were identified in myometrium and leiomyoma tissue samples, with hierarchical clustering analysis revealing tissue-type-specific CDK8 submodule occupancy in myometrium and leiomyoma (Supplementary Fig. 7e). Despite a difference in occupancy profiles between myometrium and leiomyoma samples, a global loss of CDK8 chromatin binding was not observed. However, 18% ($n = 5409/30,697$) of CDK8 submodule binding sites show changes in binding affinity in leiomyoma tissue samples as compared to myometrium (Fig. 5a and Supplementary Fig. 7b, d). A positive correlation is observed in CDK8 and MED12 occupancy changes. However, unlike AP-1 loss of binding, submodule differential binding in leiomyoma samples was characterized by nearly equal numbers of increased and decreased binding sites, with 2519 sites of submodule enrichment and 2890 sites of submodule depletion. Also, changes in CDK8 submodule binding are positively correlated with changes in H3K27Ac signal at enhancers (Fig. 5b, c). This suggests that changes in CDK8 submodule chromatin occupancy reflect an altered enhancer landscape in diseased tissues rather than a global loss of cyclin C-CDK8 binding at sites of RNAPII-mediated transcription as a result of MED12 exon 2 mutations.

Given the high correlation between H3K27 acetylation and CDK8 submodule binding at enhancers, the overlap between AP-1 and the CDK8 submodule at differential AP-1 enhancer binding sites was investigated. All JUN and FOS differentially bound sites were also co-bound by CDK8 and MED12 (Fig. 5d, e). Significantly, changes in CDK8 and MED12 binding affinities are positively correlated with changes in JUN and FOS binding at differentially bound JUN and FOS sites. AP-1 complex and CDK8 submodule co-occupancy is observed at altered enhancer regions of *FN1* and *ETS2* (Fig. 5f, g). Together, these results establish that global loss of CDK8 chromatin binding is not an epigenetic feature of *MED12* mutant leiomyomas. Rather, alterations in enhancer chromatin state, characterized by changes in H3K27Ac signal, promoter contact strength, and altered AP-1 complex/CDK8 submodule chromatin binding, together may account for dysregulated gene expression in uterine leiomyomas.

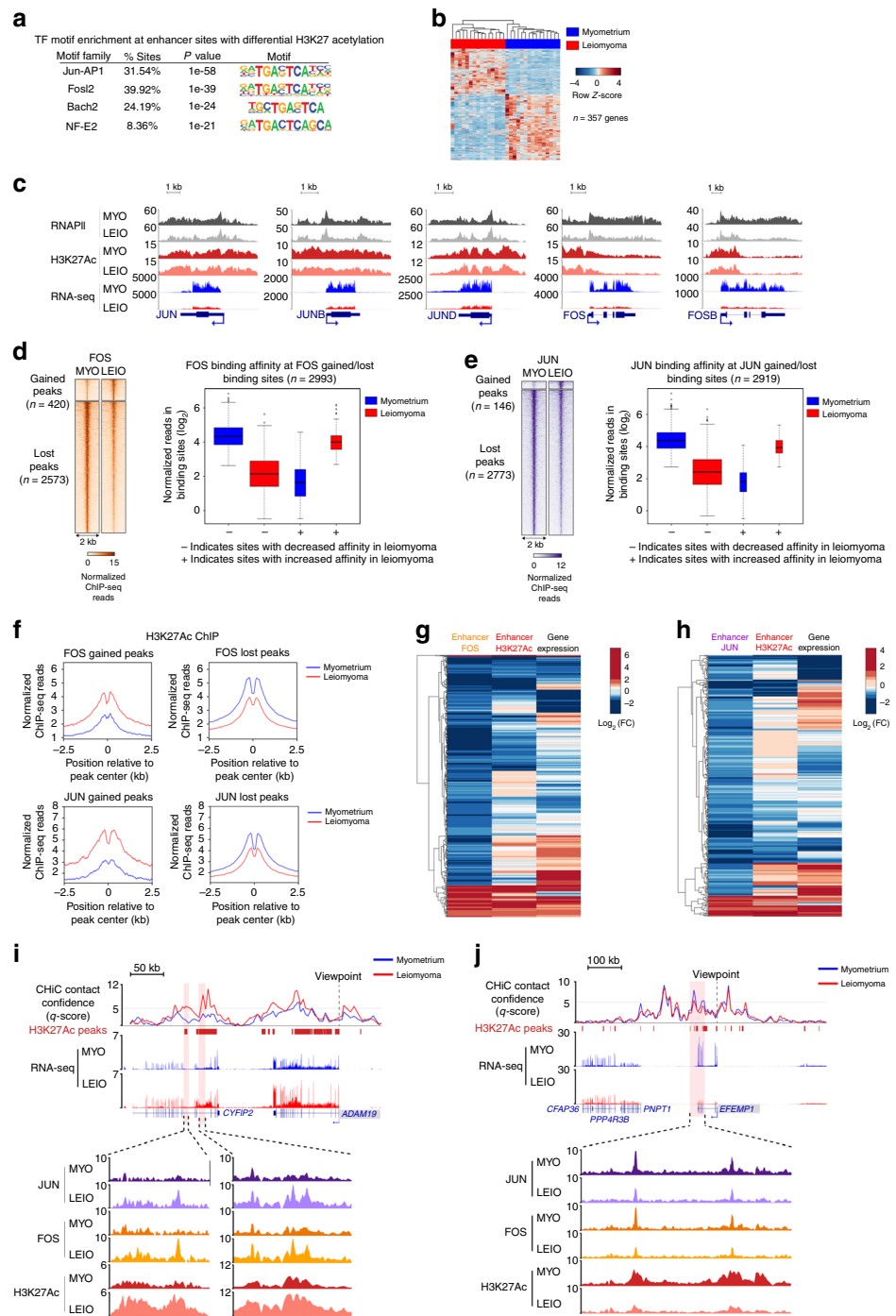

**Fig. 4 AP-1 expression and chromatin occupancy is perturbed in leiomyomas. a** Top identified transcription factor motifs (*P* value ranking, binomial) enriched in enhancer regions with differential H3K27Ac signal between myometrium and leiomyoma. **b** Heat map of differentially expressed genes with a promoter-proximal AP-1 binding site. Gene expression levels relative to the mean expression are shown as row *Z*-scores. **c** H3K27Ac, RNAPII ChIP-seq, and RNA-seq genomic tracks for AP-1 subunits *JUN, JUNB, JUND, FOS*, and *FOSB*. **d, e** Heat maps of normalized FOS (**d**) and JUN (**e**) ChIP-seq reads at differentially bound sites in myometrium vs. leiomyoma (left panels). Heat maps represent average signal of five biological replicates. Right panels show box and whisker plots of normalized reads at differentially bound FOS (**d**) and JUN (**e**) sites in myometrium and leiomyoma. Plots of sites with depleted (−) and enriched (+) FOS (**d**) and JUN (**e**) signal are shown separately. Centerline represents the median, bounds of box represent the interquartile range (IQR) and whiskers are 1.5 × IQR. **f** Tag density plots of H3K27Ac ChIP-seq enrichment profiles at FOS (top panels) and JUN (bottom panels) gained and lost binding sites in myometrium and leiomyoma. **g, h** Heat maps of changes in gene expression, enhancer H3K27Ac, and enhancer FOS (**g**) or JUN (**h**) binding associated with differentially expressed genes. Genes are hierarchically clustered by similarity in profiles of differential gene expression and FOS (*n* = 538) or JUN (*n* = 482) binding. **i, j** Differential enhancer AP-1 enrichment at *ADAM19* (**i**) and *EFEMP1* (**j**) genomic loci. Confidence scores (CHiCAGO *q*-scores) are shown for myometrium and leiomyoma. In addition, RNA-seq genomic tracks are shown. ChIP-seq signal for H3K27Ac, FOS, and JUN in enhancer regions with differential FOS and JUN binding are highlighted (bottom zoomed in insert). Differential enhancer AP-1 binding occurs at enhancers with altered (**i**) and unaltered (**j**) enhancer-promoter contacts.

**AP-1 loss leads to enhancer changes and ECM dysregulation**. The identification of direct AP-1 target genes among dysregulated genes in leiomyoma tissue samples, in concert with the down-regulation in gene expression of AP-1 subunits and the depletion of JUN and FOS on chromatin, together suggest that AP-1 may be an important protein complex in myometrium gene regulation whose down-regulation may play a role in leiomyoma disease pathogenesis. To test this, *JUN* and *FOS* family of genes were simultaneously silenced in primary human uterine smooth muscle cells (HUtSMC) by CRISPR/Cas9-mediated protein depletion (Supplementary Fig. 8a). RNA-seq of AP-1-depleted biological replicate samples showed reproducible results and segregation dependent on AP-1 depletion or control (Fig. 6a). Importantly, 1894 genes were differentially expressed in AP-1-depleted cells, with gene ontology analysis revealing changes in expression of genes involved in ECM organization (Fig. 6b, c). These results were further complemented by AP-1 subunit gene silencing using lentiviral short hairpin RNAs, followed by RNA-seq in myometrium primary cells obtained from fresh patient tissue samples (Supplementary Fig. 8b–d). These results from parallel studies demonstrate that AP-1 loss in uterine muscles cells leads to large-scale changes in gene expression as also seen in leiomyomas.

The extensive overlap between differentially acetylated regions and sites of altered AP-1 occupancy observed in leiomyoma tissue samples suggests that AP-1 may play a role in enhancer malfunction in leiomyomas by directing acetylation states at distal sites. To test this, H3K27Ac ChIP-seq was performed on AP-1-depleted HUtSMCs. Three thousand seven hundred and twenty-seven differentially acetylated regions were identified in AP-1-depleted cells as compared to negative controls (Fig. 6d). Changes in H3K27 acetylation patterns of AP-1-depleted cells relative to negative controls were strikingly similar to acetylation changes observed in leiomyoma tissues as compared to myometrium. Similarly to differential H3K27 acetylation patterns observed in leiomyoma tissue samples, nearly equal proportions of sites showed a gain or loss in H3K27Ac signal. In addition, the majority of differentially acetylated regions were at promoter-distal sites, with a higher than expected number of differentially acetylated intergenic regions (Fig. 6e, $\chi^2(4) = 346.7$, $p < 0.05$). Motif enrichment analysis also revealed almost exclusively AP-1 motifs as the top enriched motifs at differential H3K27Ac regions, with AP-1 motifs enriched at H3K27Ac peak centers (Fig. 6f).

The similarity in gene expression and H3K27Ac profiles between AP-1-depleted cells and leiomyoma tissue samples strongly suggest that a significant part of the transcriptomic and epigenomic changes seen in uterine fibroids may be directly linked to the loss in AP-1 gene expression. Taken together, these results suggest an important role played by AP-1 in uterine muscle cell enhancer regulation, the perturbation of which results in significant changes in enhancer regulation and gene expression, including the dysregulation of ECM-associated genes, a hallmark of uterine leiomyoma disease pathogenesis.

## Discussion

Epigenomic profiling of blood disorders directly from patient samples has provided extensive insights into the pathogenesis of these diseases[36]. However, epigenomic characterization of solid tumors has remained challenging, necessitating the digestion of tumors and culturing of primary cells before profiling. Using modified methods adapted for use with fresh tissue samples, we provide an extensive transcriptomic and epigenomic character-ization of normal myometrium and *MED12* mutant leiomyoma tissues, thereby yielding a near-faithful snapshot of the differences between myometrium and leiomyoma tissues under native conditions. These methods should provide a reproducible approach for the characterization of solid tumors without the need for primary cell culture.

Through comprehensive profiling of transcribed genes in myometrium and leiomyoma, we demonstrate two distinct gene expression profiles for myometrium and leiomyoma samples. This implies a consistent mechanism of disease pathogenesis in *MED12* mutant uterine fibroids, which may be driven by the mutation. The predominant feature revealed by gene ontology analysis is a per-turbation of genes that are known to be important in the regula-tion of key biological processes in leiomyoma disease pathology such as the ECM. However, although a third of all known ECM-associated genes are dysregulated in leiomyomas, this only accounts for ~5.5% of all differentially expressed genes identified in the leiomyoma transcriptome. Our analysis also identifies other possible biological processes that are perturbed in leiomyomas, which include the regulation of hormone levels and the regulation of cell secretion. It will be key going forward to delineate expres-sion changes in genes that are important in leiomyoma patho-genesis from possible non-causal secondary effects.

This integrative study reveals perturbed enhancer regulatory mechanisms that may be involved in uterine fibroid pathobiology. Modified H3K27Ac signal enrichment at enhancers as well as altered promoter contacts are identified in uterine leiomyomas. While the significance of altered contacts in the disease state is unclear, dynamic contacts have previously been demonstrated to be important chromatin architecture rewiring events in cell dif-ferentiation and may therefore be a significant rewiring event in leiomyomas[19,20].

A survey of enhancers associated with differential gene expression identified three classes of remodeled enhancers on the basis of changing H3K27 acetylation and altered promoter con-tact strength, of which enhancer regions with stable contacts and differential acetylation were the predominant class of altered enhancers. This is consistent with previous studies showing the lineage-defining characteristics of most promoter contacts, hence their relative stability[31]. In addition to a significant overlap between genomic regions forming enhancer-promoter contacts and regions with H3K27 acetylation, some promoter contacts that do not overlap H3K27-acetylated regions were also observed. Previous studies have identified promoter-distal regions with nucleosomes containing histone H3 globular domains that are acetylated at lysine residues 64 (H3K64Ac) and 122 (H3K122Ac), but not at H3K27[37,38]. Promoter contacts that do not overlap H3K27-acetylated regions in leiomyoma may be indicative of H3K122- and H3K64-acetylated enhancers and further studies to determine the function of these distinct regions will be required. Promoter contacts that do not overlap with H3K27Ac signal may also be indicative of negative regulators such as silencers, which are associated with a different epigenetic profile from enhancers marked with H3K27Ac[39].

Differential enhancer usage, whereby two or more enhancers associated with the same gene undergo inversely related chan-ges in H3K27Ac signal and promoter contact strength, was observed in enhancers associated with differentially expressed genes in leiomyomas. This mechanism may allow for the alternation of more densely occupied/stronger enhancers with weaker ones and vice versa, resulting in changes in gene expression. It may also be a mechanism by which different groups of transcription factors and cofactors are brought into contact with promoters to modulate gene transcription in uterine leiomyomas. Taken together, the observations of alterations in H3K27Ac signal at distal sites, altered promoter contacts, and the identification of differential enhancer usage in uterine leiomyomas all suggest that in addition to important promoter-proximal gene transcription regulatory events,

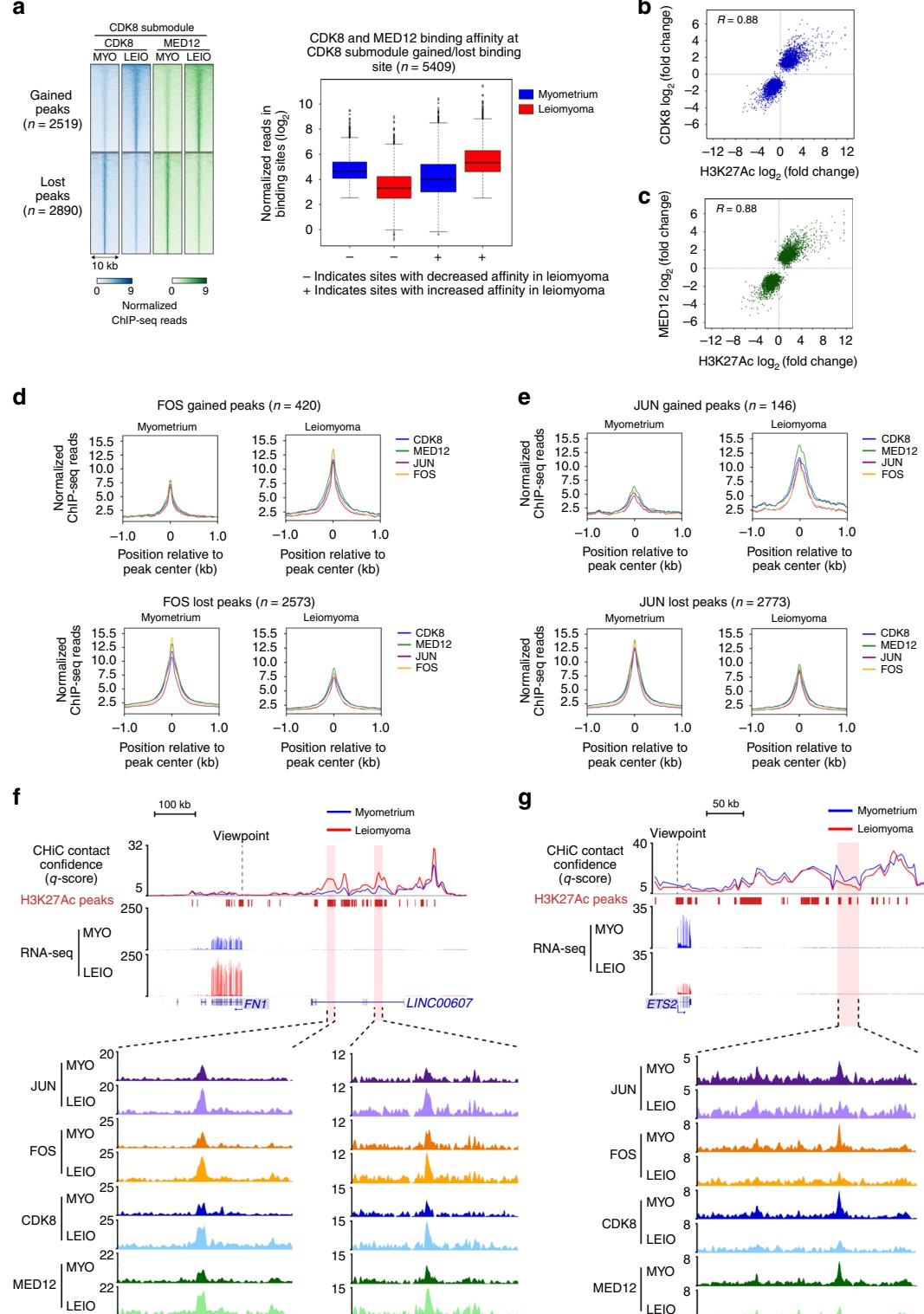

**Fig. 5 CDK8 subcomplex chromatin occupancy correlates with changes in enhancer acetylation. a** Heat maps of normalized CDK8 and MED12 ChIP-seq reads at differentially bound sites in myometrium vs. leiomyoma (left panel). Heat maps represent average signal of five biological replicates. Right panel shows box and whisker plot of normalized reads at CDK8 and MED12 co-bound sites that show enrichment or depletion of both factors. Plots of sites with depleted (−) and enriched (+) CDK8 submodule signal are shown separately. Centerline represents the median, bounds of box represent the interquartile range (IQR) and whiskers are 1.5 × IQR. **b**, **c** Scatter plots of fold change in H3K27Ac signal against fold change in CDK8 (**b**) and MED12 (**c**) at enhancers with enriched or depleted CDK8 and MED12 binding. *R* represents the Pearson's correlation coefficient. **d**, **e** Tag density plots of CDK8, MED12, JUN, and FOS ChIP-seq enrichment profiles at FOS (**d**) and JUN (**e**) enhancer binding sites in myometrium and leiomyoma. **f**, **g** AP-1 subunits (JUN and FOS) and CDK8 submodule (CDK8 and MED12) binding profiles at *FN1* (**f**) and *ETS2* (**g**) associated genomic loci. Myometrium and leiomyoma enhancer-promoter contacts and RNA-seq genomic tracks are also shown.

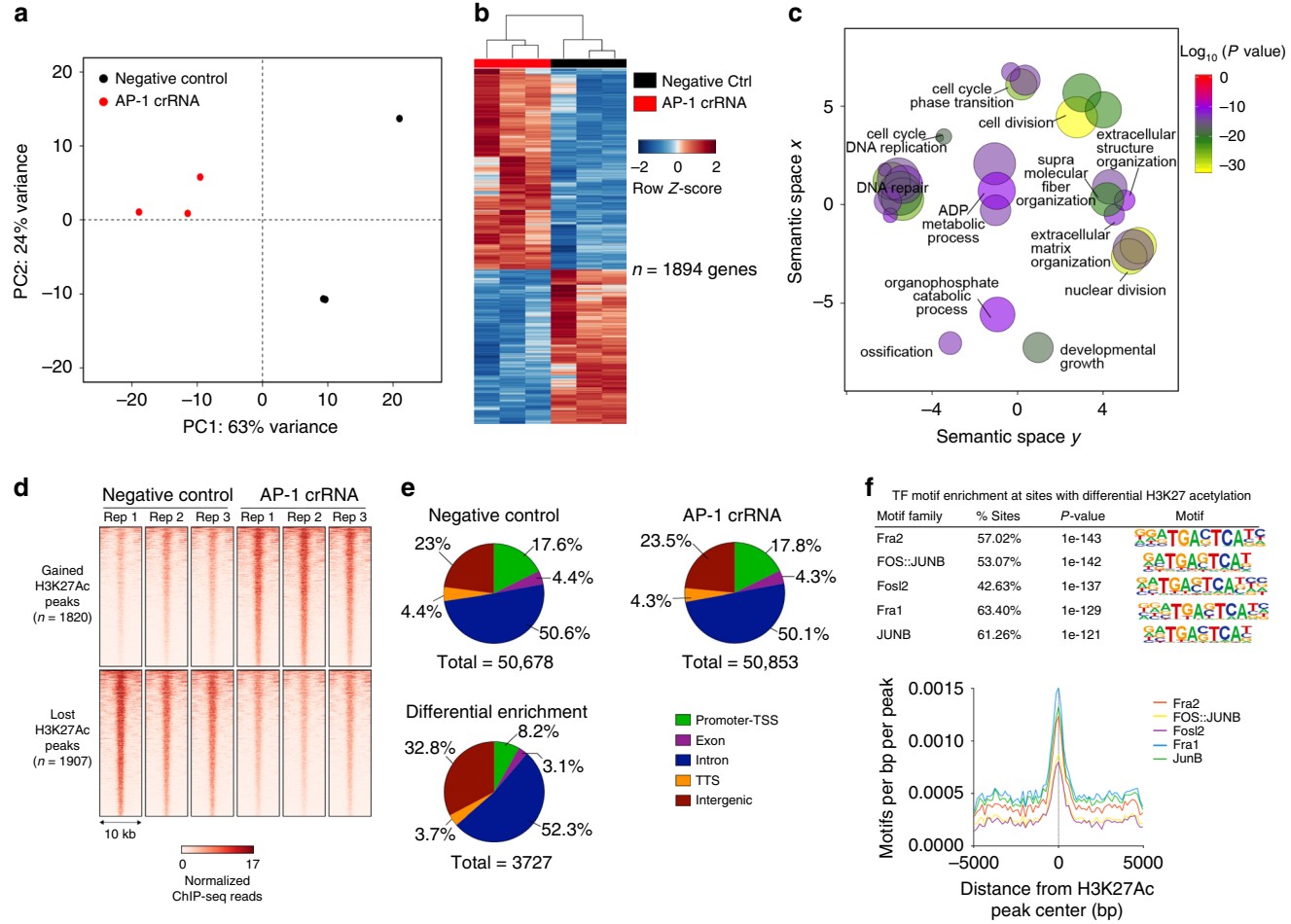

**Fig. 6 AP-1 loss leads to enhancer architecture changes and ECM gene dysregulation. a** Principal component analysis plot of the top 500 most variable genes in AP-1-depleted and negative control HUtSMCs. First two principal components are plotted. **b** Hierarchically clustered heat map of all differentially expressed genes in AP-1-depleted HUtSMCs. Gene expression levels relative to the mean expression are shown as row Z-scores. **c** Scatter plot of confidence scores for enriched gene ontologies associated with differentially expressed genes in AP-1-depleted HUtSMCs, with ontologies clustered by functional similarity in the semantic space. **d** Heat map of normalized H3K27Ac ChIP-seq reads at differentially acetylated regions in negative control vs. AP-1 depleted HUtSMCs. Signal for each replicate is shown. **e** Pie charts of H3K27Ac signal distribution at genomic loci in negative control HUtSMC (top left), AP-1-depleted HUtSMC (top right), and at differentially acetylated regions in HUtSMCs (bottom left). **f** Top identified transcription factor motifs (P value ranking, binomial) enriched at sites with differential H3K27Ac signal between negative control and AP-1-depleted HUtSMCs (top panel). Lower panel shows location of AP-1 motifs relative to peak center for all differentially acetylated regions.

enhancer malfunction is also an important mechanism of gene dysregulation in uterine leiomyomas.

The AP-1 complex is a transcription factor composed of homodimers or heterodimers from the JUN, FOS, and ATF families of transcription factors[40]. The dimeric composition of AP-1 determines which AP-1-responsive genes are regulated and whether the resulting AP-1 complex is gene activating or repressive, either directly or through the recruitment of gene-activating or repressive chromatin-modifying complexes[40,41]. Also, AP-1 has been previously implicated in the development of many fibrotic diseases[42–44].

Down-regulation of *JUN* and *FOS* mRNA levels has been observed in uterine fibroids[10,32]. However, the significance of this observation and the mechanisms by which AP-1 down-regulation led to changes in transcription were not clearly understood. In this study, a loss of *JUN* and *FOS* gene expression leads to a concomitant decrease in AP-1 recruitment at enhancer sites, which is correlated with decreased H3K27Ac signal. Enhancers associated with differentially expressed genes also show perturbations in AP-1 occupancy. In addition, loss of AP-1 chromatin occupancy at enhancers partially overlaps with changes in promoter contact strength. This is consistent with the described role of AP-1 in mediating regulatory chromatin contacts and suggests that loss of AP-1 gene expression and chromatin occupancy may drive the remodeling of enhancer architecture in leiomyomas[45]. CRISPR/Cas9-mediated silencing of AP-1 subunits in HUtSMCs demonstrates that loss of AP-1 can indeed cause the large-scale gene expression and enhancer acetylation changes seen in leiomyoma tissue samples. As such, AP-1 may serve as a master regulator of gene expression in uterine muscle cells, the loss of which could explain the broad transcriptional changes seen in leiomyoma disease pathogenesis.

Given the interest in AP-1 transcription factor activity at enhancers as well as the implication of AP-1 in multiple fibrotic diseases, future studies on AP-1-dependent enhancer malfunction in other fibrotic diseases will be of great interest[41,45–47]. In addition, down-regulation of AP-1 is also observed across leiomyomas of different subtypes[9]. Given the role of AP-1 in widespread enhancer malfunction identified in this study, ongoing work will seek to determine if similar alterations in enhancer architecture

are also observed in leiomyomas of non-*MED12* mutant origins, with AP-1 providing a possible unifying mechanism of gene dysregulation in uterine leiomyomas of differing subtypes.

Despite multiple studies showing decreased expression of AP-1 subunits in fibroids, a mechanism for AP-1 down-regulation remains unknown. The loss in expression of a significant number of subunits would suggest a unifying mode of regulation for AP-1 family genes that is perturbed in uterine fibroids; however, we did not find a singular enhancer or group of enhancers that jointly regulates AP-1 genes. Studies to further elucidate the mechanisms that result in the repression of AP-1 subunit gene expression in leiomyomas will be of great importance in increasing our understanding of leiomyoma disease pathogenesis.

Although all *JUN* and *FOS* family genes are down-regulated in uterine fibroids, it is unclear if the loss in expression of all AP-1 genes is necessary to cause the enhancer defects observed in leiomyoma cells. Given that incomplete CRISPR/Cas9-mediated depletion of *JUN* family genes in this study still results in enhancer malfunction, it is possible that a loss of a subset of AP-1 genes results in a combinatorial shift in *JUN* binding partners. Alternatively, a more complete depletion of AP-1 subunits may result in the observation of greater enhancer malfunction in primary cells that more closely recapitulates changes seen in leiomyomas. Further studies will be necessary to clarify the role of different AP-1 subunits in this disease.

Previous studies have implicated aberrant DNA methylation as an important event in uterine leiomyoma pathogenesis[48,49]. Recent work suggests that DNA methylation in leiomyomas is mediated by the long non-coding RNA H19, which regulates *TET3* expression[50]. Consistent with previous reports, we observed an increase in *H19* expression in leiomyoma tissue samples relative to myometrium. In addition, we observed increased gene expression of *TET1*, *TET3*, *DNMT1*, and *DNMT3a*. Interestingly, an increase in *H19* expression is seen upon AP-1 depletion in primary uterine smooth muscle cells. This result suggests that AP-1 plays a role, directly or indirectly, in regulating *H19* expression and further emphasizes the important role of AP-1 in this disease.

The mechanism by which exon 2 mutations in *MED12* lead to leiomyomas is not well understood. Studies have suggested that *MED12* mutations may affect transcription either through the disruption of CDK8 kinase activity or through decreased binding affinity of cyclin C and CDK8 at sites of active transcription[35]. Our work demonstrates that while CDK8 binding is altered in leiomyoma tissue samples, characterized by both a loss of binding affinity at a subset of sites and a recruitment of CDK8 at new sites, the observed change is not consistent with a global loss of CDK8 recruitment that would result from a mutation-dependent MED12-cyclin C-CDK8 interaction defect. Rather, our results show that changes in the CDK8 submodule cistrome are positively correlated with altered H3K27Ac, suggesting that altered CDK8 binding is more consistent with general changes in the enhancer landscape of leiomyoma cells. This aligns with recently published data in heterologous cell lines as well as leiomyoma tissue samples, which demonstrate that the presence of MED13 is a stabilizing protein in the submodule that compensates for any *MED12* mutant-dependent interaction defect[51,52]. However, while MED13 may stabilize cyclin C-CDK8 chromatin binding in the presence of a mutation in MED12, it cannot compensate for the loss in kinase activity of the submodule. Therefore, *MED12* mutation-driven leiomyoma disease pathogenesis is more likely a result of CDK8 kinase activity defects, and not cyclin C-CDK8 binding defects, and future work will be required to identify the kinase substrate of the CDK8 submodule.

In summary, this work highlights enhancer malfunction as an important mechanism of gene dysregulation in uterine

leiomyomas. Altered enhancer architecture, consisting of changes in histone acetylation, differential enhancer usage, and promoter contact rewiring, plays a key role in leiomyoma differential gene expression. In addition, AP-1 is shown to be an important complex in the regulation of ECM-associated biological pathways in uterine muscle tissue. This study establishes the loss of *FOS* and *JUN* gene expression in leiomyoma tissues, along with the resulting depletion of AP-1 chromatin occupancy, as likely significant events in leiomyoma disease pathogenesis.

## Methods

**Collection of uterine tissue samples.** Tumor samples were collected from 15 premenopausal women undergoing hysterectomies as a treatment for uterine fibroids. Normal, adjacent myometrium tissue samples were also collected from hysterectomies. Sanger sequencing of PCR-amplified genomic DNA extracted from leiomyomas and matched myometrium was used to positively identify *MED12* mutant leiomyoma samples with either a G44D or G44S mutation. Tissue samples were either immediately cryopreserved at −80 °C, processed for high-resolution ChIP-seq, RNA-seq, and promoter capture Hi-C, or processed for primary cell culture. Tissue from all 15 patients was used in RNA-seq experiments and tissue from five of the patients was used for ChIP-seq and capture Hi-C experiments. All surgeries were performed at Northwestern University Prentice Women's Hospital and all patients gave informed consent for their participation in this study. The study was carried out in accordance with a Northwestern University institutional review board-approved protocol. Samples were collected from hysterectomies performed on eight black women, three white women, and one latino woman. The race/ethnicities of three women were unknown. Patient ages ranged from 38 to 51 years with a mean of 46.1 years.

**Tissue digestion and primary cell culture.** Myometrium and leiomyoma tissue were cut into 5 mm³ pieces and digested with shaking (100 r.p.m.) at 37 °C for 5 h in collagenase digestion buffer (phenol red-free Hank's balanced salt solution [Thermo Fisher, cat. # 14025], 2500 KU deoxyribonuclease I [Sigma-Aldrich, cat. # D5025], 1.5 mg/ml collagenase [Sigma-Aldrich, cat. # C0130], 3 mM CaCl₂) at a ratio of 1 part w/v tissue with five parts digestion buffer. Digested tissue was filtered with sterile gauze sponges and centrifuged for 5 min at $400 \times g$. Cells were resuspended in smooth muscle cell culture media (SmGM-2 smooth muscle cell growth medium BulletKit [Lonza, cat. # CC-3182], 1% penicillin–streptomycin), filtered with a 70 μm cell strainer, plated, and grown at 37 °C in a humidified cell culture incubator containing 5% CO₂. Primary cells were used in experiments after two passages.

**Tissue RNA extraction and sequencing.** Thirty milligram of myometrium and leiomyoma tissue was cryopulverized into a fine powder and homogenized in Qiazol lysis reagent using a Kinematica Polytron PT2100 homogenizer. Total RNA was isolated using Qiagen miRNeasy Mini Kit according to the manufacturer's instructions (Qiagen, cat. # 217004). After RNA elution, an in-solution DNase treatment was performed to completely digest genomic DNA (Qiagen, cat. # 79254). RNA quality was verified using the Bioanalyzer Eukaryote Total RNA 600 Nano Kit (Agilent Technologies, cat. # 5067). Purified RNA samples had a mean RNA integrity number of 8.0. RNA was prepared for sequencing using the KAPA-stranded RNA-seq with RiboErase Kit according to the manufacturer's instructions (Kapa Biosystems, cat. # KK8483). Paired-end sequencing of all RNA libraries was performed on the Illumina NextSeq 500 Platform.

**Tissue preparation for ChIP and capture Hi-C.** Myometrium and leiomyoma tissue samples were cryopulverized into a fine powder using a Covaris cryoPREP CP02 impactor followed by a mortar and pestle. Samples were immediately formaldehyde cross-linked (1% formaldehyde in phosphate-buffered saline) for 10 min at room temperature with rotation and then quenched with 0.15 M glycine for 10 min at room temperature with rotation. Cross-linked tissue was recovered by centrifugation at $1000 \times g$ at 4 °C for 5 min and the supernatant discarded. Two washes in phosphate-buffered saline containing 1× Protease inhibitor cocktail (Roche, cat. # 4693132001) was performed and tissue recovered by centrifugation at $1000 \times g$ at 4 °C for 5 min. Cross-linked tissue was then either frozen at −80 °C or immediately used in ChIP or CHi-C experiments.

**Tissue chromatin immunoprecipitation and sequencing.** Chromatin immunoprecipitation was adapted from previously described methods[53]. Cross-linked myometrium and leiomyoma tissue samples were dounce homogenized in ChIP lysis buffer 1 (50 mM HEPES-KOH [pH 7.6], 140 mM NaCl, 1 mM EDTA, 10% glycerol, 0.5% IGEPAL CA-630, 0.25% Triton X-100, 1× Protease inhibitor cocktail [Roche, cat. # 4693132001]), followed by end-over-end rotation at 4 °C for 15 min and then centrifugation for 5 min at $1000 \times g$, also at 4 °C. Samples were resuspended in ChIP lysis buffer 2 (10 mM Tris-HCl [pH 8.0], 200 mM NaCl, 1 mM EDTA, 0.5 mM EGTA, 1× Protease inhibitor cocktail [Roche, cat. # 4693132001]),

followed again by end-over-end rotation at 4 °C for 15 min and sample recovery by centrifugation at 1000 × g at 4 °C for 5 min. Samples were then resuspended in MNase digestion buffer (20 mM Tris-HCl [pH 8.0], 150 mM NaCl, 2 mM EDTA, 1% Triton X-100, 0.1% sodium dodecyl sulfate [SDS], 125 U MNase [Worthington, cat. # LS004798], 1× Protease inhibitor cocktail [Roche, cat. # 4693132001]), and incubated at 37 °C until 75% mono-nucleosomal chromatin profile is observable by gel electrophoresis of purified MNase-digested DNA (~5–7 min). Digestion was quenched with MNase quenching buffer (10 mM EDTA, 20 mM EGTA), after which samples were briefly sonicated in an ice water bath (Misonix, setting 6 [~6 W power output], three cycles of 15 s on and 45 s off). Collagen and cell debris was removed by centrifugation at 20,000 × g at 4 °C for 20 min. Solubilized chromatin concentration was measured by the BCA assay and ~400 μg of chromatin and 3 μg of antibody was used for overnight immunoprecipitation of histones at 4 °C with end-over-end rotation, while 600 μg of chromatin and 5 μg of antibody was used for overnight immunoprecipation of factors. Antibodies against H3K27Ac (Active Motif, cat. # 39685), RNAPII, FOS (Millipore, cat. #s 05-952, 06-341, respectively), JUN (Abcam, cat. # ab31419), and CDK8 and MED12 (Bethyl, cat. #s A302-500A, A300-774A, respectively) were used in pulldowns. Ten microliters of protein G magnetic beads (Thermo Fisher, cat. # 10004D) per μg of antibody were added and samples were incubated at 4 °C for an additional 3 h. Immunoprecipitated samples bound to beads were recovered using a magnetic rack (Thermo Fisher, cat. # 12321D), followed by 5 washes with ChIP-RIPA wash buffer (50 mM HEPES-KOH [pH 7.6], 500 mM LiCl, 1 mM EDTA, 1% IGEPAL CA-630, 0.7% sodium deoxycholate) and once with NaCl containing TE buffer (10 mM Tris-HCl [pH 8.0], 1 mM EDTA, 50 mM NaCl). DNA was eluted off the beads twice with 50 μl ChIP elution buffer (0.1 M NaHCO$_3$, 1% SDS) at 65 °C for 20 min with shaking. Cross-links were reversed with 400 mM NaCl for 12–16 h at 65 °C, followed by RNase A treatment at 37 °C for 1 h. This was followed by 2 h at 55 °C with 40 μg of proteinase K (Thermo Fisher, cat. # 25530015) supplemented with 16.5 mM EDTA and 66 mM Tris-HCl (pH 8.0). Reverse cross-linked DNA was isolated by phenol–chloroform extraction. KAPA Hyper Prep Kit (Kapa Biosystems, cat. # KK8502) was used for end repair, A tailing, and adapter ligation with TruSeq index adapters, all according to the manufacturer's instructions. A 1.1×–1.8× AMPure XP bead (Beckman Coulter, cat. # A63881) size selection was carried out after adapter ligation to enrich for sub-nucleosomal DNA fragments, followed by 10–12 cycles of PCR amplification. For H3K27Ac ChIP, a 0.6×–1.0× AMPure XP bead size selection was performed to enrich for mono-nucleosomal DNA fragments. Single-end sequencing of all ChIP libraries was performed on the Illumina NextSeq 500 platform.

**Tissue promoter capture Hi-C**. Promoter capture Hi-C was adapted from previously described methods[18,54–57]. Cross-linked myometrium and leiomyoma tissue samples were dounce homogenized in permeabilization buffer (10 mM Tris-HCl [pH 8.0], 10 mM NaCl, 0.2% IGEPAL CA-630, 1× Protease Inhibitor cocktail) followed by end-over-end rotation at 4 °C for 15 min, after which sample pellet was recovered by centrifugation at 4 °C. Samples were washed in 1.2× NEBuffer2 (cat. # B7002) at 4 °C for 15 min with end-over-end rotation, followed by centrifugation at 4 °C. Samples were then resuspended in 1.2× NEBuffer2 containing 0.2% SDS buffer and incubated at 37 °C for 1 h with end-over-end rotation, followed by the addition of 1.8% Triton X-100 at 37 °C for an additional hour. The resulting chromatin was digested with 400 U of HindIII restriction enzyme (NEB cat. # R0104) for 12–16 h at 37 °C with end-over-end rotation. This was followed by DNA polymerase I-mediated (50 U [NEB, cat. # M0210]) labeling of DNA ends with dNTPs (28.4 μM D-desthiobiotin-7-dATP [Jenna Bioscience, cat. # NU-835-Desthiobio], 28.4 μM dCTP [Invitrogen, cat. # 18253-013], 28.4 μM dGTP [Invitrogen, cat. # 18254-011], 28.4 μM dTTP [Invitrogen, cat. # 18255-018]), for 3 h at 37 °C. Samples were then incubated at room temperature for 4 h with end-over-end rotation in proximity ligation mix (1× T4 DNA Ligase buffer [NEB, cat. # B0202], 1× bovine serum albumin [NEB, cat. # B9001], 2000 U of T4 DNA ligase [NEB, cat. # M0202]), followed by the addition of SDS to a final concentration of 1%. Samples were then treated with RNase A at 37 °C for 1 h and proteinase K at 55 °C for 2 h. Cross-links were reversed with 400 mM NaCl for 12–16 h at 65 °C. Labeled and proximity ligated DNA was isolated by phenol–chloroform extraction and sheared to a size of 300–500 bp using a Covaris S2 focused ultrasonicator, followed by a 0.5×–0.9× AMPure XP bead (Beckman Coulter, cat. # A63881/ A63882) size selection. Biotin was removed from unligated ends with T4 DNA polymerase mix (1× NEBuffer2, 0.05 mM dATP, 0.05 mM dGTP, 15 U of T4 DNA polymerase (NEB, M0203)) at 20 °C for 4 h, followed by streptavidin bead-based biotin pull-down (Thermo Fisher Scientific, cat. # 65001) in 1× streptavidin binding buffer (5 mM Tris-HCl [pH 7.5], 0.5 mM EDTA, 1 M NaCl). Beads were washed five times in streptavidin Tween washing buffer (5 mM Tris-HCl [pH 7.5], 0.5 mM EDTA, 1 M NaCl, 0.05% Tween-20) and DNA was eluted off the beads twice with 50 μl biotin elution buffer (10 mM biotin solution [biotin solubilized in 0.1 N NaOH], 20 mM Tris-HCl [pH 8.0]) at room temperature for 30 min, followed by a 1× AMPure XP bead cleanup. KAPA Hyper Prep Kit (Kapa Biosystems, cat. # KK8502) was used for end repair, A tailing, adapter ligation with TruSeq index adapters, and 12 cycles of PCR amplification, all according to the manufacturer's instructions. Previously described custom-designed biotinylated 120-mer RNA baits targeting Ensembl promoters of protein-coding and non-coding transcripts were used to capture promoter associated biotinylated and adapter ligated Hi-C

DNA from myometrium and leiomyoma samples using Agilent SureSelect XT Kit (Agilent Technologies, cat. # G9611A) according to the manufacturer's instructions. TS universal blocker and TS index-specific blocker (IDT, xGen standard blocking oligos) were used in place of the SureSelect ILM Indexing Block 3. Six cycles of post-capture, on-bead PCR amplification was performed and paired-end sequencing of all CHi-C libraries was performed on the Illumina NextSeq 500 platform.

**RNAi-mediated silencing in primary cells and RNA-seq**. Short hairpin sequences targeting AP-1 subunits JUN, JUNB, JUND, FOS, and FOSB were obtained from the RNAi consortium (TRC). Short hairpin oligonucleotides (Integrated DNA Technologies) targeting AP-1 subunits and non-silencing control were cloned into a modified pLVX-IRES-mCherry vector (Clontech, cat. # 631237) with bicistronic transcription driven by the phosphoglycerate kinase (PGK) promoter instead of the cytomegalovirus immediate early (CMV IE) promoter. Lentivirus production was carried out in HEK 293 T/17 cells (ATCC, CRL-11268) with lipofectamine co-transfection of pMD2.G, psPAX2, and shRNA-containing lentiviral construct, a total of 18 μg DNA at a ratio of 3:2:1, according to manufacturer's instructions (Thermo Fisher, cat. # 11668019). Virus was concentrated 10-fold using polyethylene glycol according to manufacturer's instructions (Takara Bio, cat. # 631231). $1.0 \times 10^6$ myometrium primary cells collected from each patient ($n = 2$ biological replicates) and cultured in 10 cm$^2$ dishes with SmGM-2 muscle cell media, were each transduced with 100 μl of either non-silencing control or AP-1 gene silencing concentrated virus and 6 μg/ml of polybrene for 18 h. This amount of virus was experimentally determined to be the minimum amount required to give >80% transduction efficiency as determined by mCherry fluorescence. Fresh SmGM-2 muscle cell media was then added and cells were allowed to grow for 3–4 days. Cells were then harvested and RNA extracted using the RNeasy Mini Kit according to manufacturer's instructions (Qiagen, cat. # 74104). Gene knockdown was confirmed by RT-qPCR and western blot of JUN, JUNB (Bethyl, cat. #s A302-958A [1:1000], A302-704A [1:1000] respectively), JUND (Abcam, cat. # ab28837 [1:500]), FOS (Thermo Fisher, cat. # MA5-15055 [1:250]), and FOSB (Active motif, cat. # 40960 [1:2000]). Uncropped western blot images are provided in Supplementary fig. 9. Purified RNA was prepared for sequencing using the KAPA-Stranded RNA-seq with RiboErase Kit according to the manufacturer's instructions (Kapa Biosystems, cat. # KK8483). Paired-end sequencing of all libraries was performed on the Illumina NextSeq 500 platform.

**CRISPR/Cas9-mediated silencing of uterine smooth muscle cells**. HUtSMCs were obtained from American Type Culture Collection (ATCC, cat. # PCS-460-011) and grown in vascular smooth muscle cell complete media (ATCC, cat. #s PCS-100-030, PCS-100-042). Cells were grown at 37 °C in a humidified cell culture incubator containing 5% CO$_2$. Media was changed every 48 h and cells were passaged at 90% confluence.

Custom CRISPR-Cas9 crRNAs targeting AP-1 subunits JUN, JUNB, JUND, FOS, and FOSB gene loci (IDT, cat. # 11-01-03-01) were combined with tracrRNA (IDT, cat. # 1072533) in a 1:1 ratio to a final concentration of 44 μM, incubated at 95 °C for 5 min and allowed to cool to room temperature. SpCas9 enzyme (IDT, cat. # 1081058) was diluted to 36 μM with Neon resuspension buffer R (Thermo Fisher, cat. # MPK1025) and combined with the crRNA:tracrRNA AP-1 mixes in a 1:1 volume ratio and incubated at room temperature for 20 min, after which electroporation enhancer (IDT, cat. # 1075916) was added to a final concentration of 1.8 μM. HUtSM cells that were 90% confluent were trypsinized for 4 min, quenched in FBS-containing media and centrifuged at 300 × g for 5 min at room temperature. Centrifuged cells were resuspended in phosphate-buffered saline and counted. Cells were then centrifuged at 300 × g for 5 min at room temperature and resuspended at a concentration of 15,000 cells/μl in Neon resuspension buffer R. 135,000 cells were combined with crRNA:tracrRNA:cas9:electroporation enhancer mix targeting either a non-silencing control or AP-1 (JUN, JUNB, JUND, FOS, and FOSB) and electroporated in a 10 μl Neon pipette tip (pulse voltage = 1,500 V, pulse width = 10 ms, # of pulses = 3) using a Neon electroporation system (Thermo Fisher, cat. # MPK5000) according to manufacturer's instructions. Electroporation of 135,000 cells in a 10 μl Neon pipette tip was repeated ten times and $1.35 \times 10^6$ electroporated cells were then cultured in smooth muscle cell complete media. Gene editing was confirmed by sequence trace decomposition[58].

**RNA extraction and sequencing of AP-1 depleted HUtSMCs**. Cells were harvested and RNA extracted using the RNeasy Mini Kit according to the manufacturer's instructions (Qiagen, cat. # 74104). Purified RNA was prepared for sequencing using the KAPA mRNA Hyper Prep Kit according to the manufacturer's instructions (Kapa Biosystems, cat. # KK8580). Paired-end sequencing of all libraries was performed on the Illumina NextSeq 500 platform.

**ChIP and sequencing of AP-1-depleted HUtSMCs**. Sixteen percent paraformaldehyde was added to full media-containing HUtSM cell plates to a final concentration of 1% paraformaldehyde and incubated at room temperature for 10 min. Cross-linked cells were then quenched with 0.125 M glycine solution for a further 10 min, followed by multiple washes in cold (stored at 4 °C) phosphate-buffered saline containing 1× protease inhibitor cocktail (Roche, cat. #

4693132001). Cells were then centrifuged for 5 min at $1000 \times g$ and then resuspended in ChIP lysis buffer 1 (50 mM HEPES-KOH [pH 7.6], 140 mM NaCl, 1 mM EDTA, 10% glycerol, 0.5% IGEPAL CA-630, 0.25% Triton X-100, 1× Protease inhibitor cocktail [Roche, cat. # 4693132001]), followed by end-over-end rotation at 4 °C for 15 min and then centrifuged for 5 min at $1000 \times g$, also at 4 °C. Samples were resuspended in ChIP lysis buffer 2 (10 mM Tris-HCl [pH 8.0], 200 mM NaCl, 1 mM EDTA, 0.5 mM EGTA, 1× Protease inhibitor cocktail [Roche, cat. # 4693132001]), followed again by end-over-end rotation at 4 °C for 15 min and sample recovery by centrifugation at $1000 \times g$ at 4 °C for 5 min. Samples were then resuspended in ChIP lysis buffer 3 (10 mM Tris-HCl [pH 7.5], 100 mM NaCl, 1 mM EDTA, 0.5 mM EGTA, 0.1% sodium deoxycholate, 0.5% sarkosyl) and sonicated in an ice water bath (Misonix, setting 6 [~6 W power output], 12 cycles of 15 s on and 45 s off). Triton X-100 was added to a final concentration of 1% and samples were centrifuged at $20,000 \times g$ at 4 °C for 20 min, with recovery of chromatin-containing supernatant. Solubilized chromatin concentration was measured by the BCA assay according to the manufacturer's instructions (Thermo Fisher, cat. # 23225). Approximately 500 μg of chromatin and 4 μg of antibody against H3K27Ac (Active motif, cat. # 39685) was used for overnight immuno-precipitation of histones at 4 °C with end-over-end rotation. 10 μl of protein G magnetic beads (Thermo Fisher, cat. # 10004D) per μg of antibody was added and samples were incubated at 4 °C for an additional 3 h. Immunoprecipitated samples bound to beads were recovered using a magnetic rack (Thermo Fisher, cat. # 12321D), followed by 5 washes with ChIP-RIPA wash buffer (50 mM HEPES-KOH [pH 7.6], 500 mM LiCl, 1 mM EDTA, 1% IGEPAL CA-630, 0.7% sodium deoxycholate) and once with NaCl containing TE buffer (10 mM Tris-HCl [pH 8.0], 1 mM EDTA, 50 mM NaCl), with magnetic recovery of immunoprecipitated samples bound to beads after each wash. DNA was eluted off the beads twice with 50 μl ChIP elution buffer. (0.1 M NaHCO3, 1% SDS) at 65 °C for 20 min with shaking. Cross-links were reversed with 300 mM NaCl for 12–16 h at 65 °C followed by 20 μg of RNase A (Worthington, cat. # LS002132) treatment at 37 °C for 1 h. This was followed by 2 h at 55 °C with 80 μg of proteinase K (Thermo Fisher, cat. # 25530015) supplemented with 16.5 mM EDTA and 66 mM Tris-HCl (pH 8.0). Reverse cross-linked DNA was isolated by phenol–chloroform extraction with phenol/chloroform/isoamyl alcohol [25:24:1] (Sigma, cat. # 77617) using 5Prime phase lock gel (Quantabio, cat. # 2302830) according to manufacturer's instructions, followed by ethanol precipitation (2× volume of 100% ethanol, 0.1× volume of 3 M sodium acetate, and 2 μl of glycogen) at −80 °C for 1 h. DNA was then pelleted by centrifugation at $20,000 \times g$ at 4 °C for 20 min, followed by one wash in with 70% ethanol and further centrifugation at $20,000 \times g$ at 4 °C for 10 min. DNA pellet was dried and resolubilized in 50 μl of elution buffer (10 mM Tris-HCl, pH 8.0–8.5). KAPA Hyper Prep Kit (Kapa Biosystems, cat. # KK8502) was used for end repair, A tailing, and adapter ligation with TruSeq index adapters, all according to the manufacturer's instructions. A $0.6 \times –0.8 \times$ AMPure XP bead (Beckman Coulter, cat. # A63881) size selection was carried out after adapter ligation to enrich for mono-nucleosomal DNA fragments from the H3K27Ac ChIP followed by 12 cycles of PCR amplification. Single-end sequencing of all ChIP libraries was performed on the Illumina NextSeq 500 platform.

**General analysis and data processing.** All ChIP, CHi-C, and RNA-seq reads in FASTQ format were aligned to the GRCh38 human genome assembly obtained from Ensembl (release 90) and all transcript annotations were also obtained from Ensembl. The R package Gviz was used to visualize browser tracks of ChIP-seq, and RNA-seq BigWig files as well as myometrium and leiomyoma CHi-C q-scores[59].

**RNA-seq data processing and analysis.** RNA-seq reads ($n = 15$ patients) were aligned to the GRCh38 human genome assembly using the STAR aligner (v2.5.3a) with default settings[60]. Uniform gene body coverage of reads was verified with the RSeQC python package (v2.6.4)[61]. RSeQC was also used to normalize BAM files obtained from STAR alignments to $1 \times 10^7$ reads when converting to wiggle format, which were then converted to BigWig files using the UCSC browser utility wigToBigWig. Counting of reads per gene was performed using featureCounts from the Subread package (v1.5.0; introns + exons count settings: -g gene_id -t gene -p -s 2, exon count settings: -g gene_id -t exon -p -s 2) and differential expression analysis of gene counts between myometrium and leiomyoma was performed with DESeq2 (v1.18.1), with the Wald test used to test for significance (P value). Multiple testing correction was applied using the BH method, with the false discovery rate (adjusted P value) controlled for at <0.01[62,63]. RNA tissue clustering and differential gene expression heat maps were generated in R using pheatmaps (cran. r-project.org). All reads (introns + exons) mapping to an annotated feature were used to compare transcriptome profiles between samples while only exonic reads were used for differential gene expression analysis. Enriched gene ontology terms were identified using Metascape and scatter plots of ontology terms were made using REVIGO[64,65]. EISA was performed as previously described[25].

**ChIP-seq data processing and analysis.** ChIP-seq reads ($n = 5$ patients) were first adapter trimmed using cutadapt (v1.15)[66]. Read quality of tissue samples and cells ($n = 3$) were then verified with FastQC (www.bioinformatics.babraham.ac.uk)[66]. Reads were then aligned to the GRCh38 human genome assembly using bowtie

(v1.2.2; settings: --best -m 1)[67]. Peak calling and motif enrichment were performed with HOMER (v4.10.3, homer.ucsd.edu; histone peak calling: -tbp 1 -F 4 -style histone, transcription factor peak calling: -tbp 1 -F 3 -style factor, co-factor peak calling: -tbp 1 -F 3 -style histone). Differential enrichment of peaks was determined using the R package DiffBind[68] (www.bioconductor.org). First, DiffBind was used to derive a consensus myometrium and leiomyoma peak set for each histone or factor ChIP from the individual HOMER-called peak sets of each biological replicate. Differential peak analysis on the consensus peak set was then performed with a peak being identified as differentially enriched or depleted at an FDR cutoff <0.05 and a fold change >|2|. Promoters are defined as regions from −1000 to +100 bp of the transcriptional start site and enhancer regions are H3K27Ac peaks that fall outside promoter regions and are further subdivided into H3K27Ac peaks found in exonic, intronic, and intergenic regions. Pearson's correlation heat maps of biological replicates were also drawn using DiffBind. Reads per genomic content normalization (RPGC, 1× normalization) of BAM files obtained from Bowtie alignments was performed with DeepTools to produce BigWig files[69]. Heat maps and profile plots of ChIP-seq peaks were also plotted using DeepTools. Motif enrichment analysis at differential H3K27Ac regions and at JUN/FOS enhancer-bound sites was carried out using the HOMER function findMotifsGenome using default parameters.

**Promoter CHi-C data processing and analysis.** Promoter capture Hi-C sequencing reads ($n = 5$ patients) were processed using the Hi-C User Pipeline (HiCUP, v0.5.10), with the bowtie aligner (v1.2.2) being used to align the reads to the GRCh38 human genome assembly[67,70]. The capture Hi-C analysis of genomic organization package (CHiCAGO) was used to score CHi-C contact strength, with HiCUP processed BAM files being used as input files for the CHiCAGO pipeline[71]. Enhancer-promoter (E-P) and promoter-promoter (P-P) interactions in myometrium and leiomyoma samples with a CHiCAGO score >5 were considered to be valid contacts. The edgR package (v3.20.9) was used to identify significantly altered contacts at an FDR <0.1[72]. Only contacts with a mean read count >10 for each tissue type were used in the edgeR analysis. Heat maps of differential contacts were generated in R using pheatmaps (cran.r-project.org). HindIII digest of DNA in CHi-C sample preparation was leveraged as a means to assign overlap between E-P contacts and H3K27-acetylated regions. H3K27Ac peaks that overlapped the same HindIII fragment as the distal other end of an E-P contact were assigned to the same HindIII fragment/region (related to Fig. 3a and Supplementary Fig. 3a).

**Reporting summary.** Further information on research design is available in the Nature Research Reporting Summary linked to this article.

## Data availability

All high-throughput sequencing data as well as source data underlying figures that support the findings of this study have been made available through the National Center for Biotechnology Information (NCBI) Gene Expression Omnibus (GEO) data repository and can be accessed via accession GSE128242. All data analysis was performed with publicly available software and is listed in the methods. All other relevant data supporting the key findings of this study are available within the article and its Supplementary Information files or from the corresponding author upon reasonable request. A reporting summary for this Article is available as a Supplementary Information file.

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

## Acknowledgements

Stacy A. Kujawa and Saurabh S. Malpani obtained consent from patients and collected tissue samples for this study. Yasuhiro Omura and Amanda L. Allred performed high-throughput sequencing of all RNA-seq, ChIP-seq, and CHi-C libraries. We thank Dr. Joseph T. Bass, Dr. Grant D. Barish, and the Feinberg School of Medicine, Division of Endocrinology, Metabolism and Molecular Medicine for providing access to the Illumina NextSeq 500 sequencing platform for this study. We would also like to thank Dr. Steven Wingett and Dr. Mikhail Spivakov for helpful discussions regarding HiCUP and CHi-CAGO analysis packages, respectively. This work was supported by the Northwestern University NUSeq Core Facility and NIH grants R21HD082781, R01HD089552, and P01HD057877.

## Author contributions

M.B.M., J.B.P., and D.C. conceived the research plan. M.B.M. designed experiments, performed research, analyzed data, and wrote the initial draft. M.B.M., J.B.P., and D.C. revised the manuscript. D.C. supervised the project and acquired funding.

## Competing interests

The authors declare no competing interests.
