## [Peer Review File · Nature Communications]

Reviewers' comments:

Reviewer #1 (Remarks to the Author):

Mayo and colleagues studied MED12 mutant uterine leiomyomas from 15 patients to initially identify differentially expressed genes in leiomyomas as compared to adjacent non-diseased myometrium. Indeed, they found an astonishing number of close to 6000 differentially regulated genes, which is about one quarter of all genes that are encoded in the human genome. Then, they performed ChIP-sequencing for the H3K27ac mark to find differentially acetylated regions in the genome. Again, they find a substantial fraction of 30% differentially acetylated regions in the genome. Using promoter capture Hi-C they find that the enhancer architecture is greatly altered in MED12-mutant uterine leiomyomas and is associated with a depletion of AP-1 occupancy on chromatin. They then establish primary myometrium cells and silence several subunits of the AP-1 complex. Indeed, gene sets are dysregulated in the knockdown conditions similar to the situation in leiomyomas. The authors conclude that AP-1 would be the driver of aberrant enhancer regulation and that this would be an important mechanism of leiomyoma pathogenesis.

It does not become fully clear from the study what the drivers of the epigenetic changes and altered enhancer-promoter interactions are that go together with reduced AP-1 occupancy. Are these changes indeed the drivers or rather passenger events in the disease?

The authors specifically analyzed leiomyomas carrying a particular mutation in the MED12 gene. This mutation has previously been described as a gain-of-function mutation which was sufficient to induce leiomyomas and genomic instability in mice (PMID:26193636). This work and that recently published on lincRNA H19 (PMID:31089260) should be regarded by the authors and the respective findings at least be discussed in the context of their own observations. In the latter paper, TET3 activity on DNA is described in the context of leiomyomas, which might contribute to the interpretation of some findings that are reported here. For example, could there be a relation between de-/methylation of DNA and AP1 activity in the context of leiomyoma (even if the AP1 motif does not contain a CpG dinucleotide)? What is the relation between de-/methylation of DNA and the binding of the Mediator complex and the regulation of PolII transcription? On page 15 the authors state that "the observation of alterations in enhancer-promoter contacts and the identification of differential enhancer usage in uterine leiomyomas establishes the remodeling of enhancer regions not only important in normal enhancer-mediated regulation of gene expression but also key in the development of disease." The evidence that is provided here does not really allow to conclude whether these alterations are causal or rather mediator/symptom for the disease. The link between the activating mutation in MED12 via H19 and TET3 could help to better explain some of the findings that are described in the current manuscript. Similarly, the authors claim (p16) that their "study describes the role of AP-1 in extracellular matrix gene regulation and the mechanisms by which decreased AP-1 expression leads to the formation of uterine leiomyomas." The authors should tone down this statement in the light of the literature described above. The interesting question should be what leads to downregulation of AP-1 in leiomyomas. The authors should discuss others' findings in the context of their own.

On page 12 the authors write that they wanted to assess whether exon2 mutations at G44 in MED12 protein would result in a loss of CDK8 submodule chromatin binding in uterine leiomyomas. This hot-spot gain-of-function mutation is the only recurrent mutation in the MED12 gene and has been observed in several malignancies. The mutation is indeed likely to alter the activity of the Mediator complex, however, this is unlikely to result in a loss of CDK8 submodule chromatin binding. Indeed, the authors observed different occupancy profiles of wildtype vs. mutant MED12, however, not a global loss of submodule chromatin binding. On the same page, they speak of a tissue type-specific CDK8 submodule occupancy in myometrium and leiomyoma (Supplementary Fig. 7e). The differences in hierarchical clustering might well be caused by different binding patterns of wt vs. mutant MED12. If true, this would be similar to some mutations (in the DNA binding-domain) in TP53 which have been shown to alter the spectrum of binding sites the respective mutant forms of p53 have compared to wt p53. On page 17, the authors write that "MED13 is a stabilizing protein in the submodule that compensates for any MED12 mutant dependent interaction defect" (citing references 61 and 62). The role mutant MED12 has in the

formation of leiomyomas speaks against a fully efficient compensation of the altered activity by MED13. A mutant dependent interaction defect would typically be observed in a tumor suppressor gene. The hot-spot nature of the mutation and the published effects mutant MED12 induces rather speak for an oncogenic function the protein acquires by the mutation. The authors should rephrase their statement.

In a follow-up study it might be interesting to investigate also patients who are wildtype for MED12 in the respective leiomyoma. This might help to even better decipher the direct mechanisms that are driven by mutant MED12 and to potentially uncover alternative mechanisms of leiomyoma development. Is loss of AP1 also prevalent in wt MED12 leiomyomas? Is the chromatin stable there?

The same hot-spot mutation in codon 12 has been found also in 1/3 of soft tissue tumors, 12% of breast cancers, and ~5% of skin cancers and other tumor entities. Overexpression of MED12 has been detected in tumors of the urinary tract, stomach cancer, soft tissue tumors, oesophageal cancer, and tumors of the large intestine, among others (Cosmic database). It will be interesting to see whether similar changes in the epigenetic landscape of these tumors exist and if similar mechanisms are relevant also in other tumor diseases.

Minor

On page 10 the authors write that Fig. 4c should show expression of JUN, FOS and ATF families of genes. The data is shown for JUN and FOS, however, not for members of the ATF family of transcription factors.

On page 20 the authors should provide a g-force when they speak of low speed centrifugation. Below they should add a temperature when they describe brief sonication of samples using a Misonix device.

On page 23 they describe the viral transduction of concentrated lentiviral particles for silencing AP-1 gene or non-silencing control. The MOI should be provided.

Which software was used to align ChIP and Chi-C reads to the genome (page 24)? The next paragraph speaks of STAR aligner for RNA-seq data.

Stefan Wiemann

Reviewer #2 (Remarks to the Author):

In this manuscript, Moyo et al. investigate epigenetic alterations (including chromatin interactions and transcription factor binding) at dysregulated transcripts in MED12 mutant fibroids. The concept of investigating chromatin interactions, which are known to be regulated by MED12, in fibroids, which are known to have dysregulated MED12, is novel and interesting. This paper is one of the first set of papers to investigate chromatin interactions in clinical samples, in this case, fibroids, and will be of interest in the field. The figures and text are well done.

I have a few major issues to discuss:

1. This paper is largely descriptive, yet the authors argue that the observations seen are a reflection of a MED12 dysregulation pattern. However, these observations are just correlations. To be able to say that the observations of altered chromatin interactions result from MED12 dysregulation, the authors would need to perform MED12 alteration (e.g. CRISPR to remove exon 2) in a cell line or two and show altered chromatin interactions as a result. The authors mention in the discussion that they are unsure whether MED13 may compensate, and that the MED12 pathogenesis may be due to CDK8 kinase activity, however, it would still be useful to establish in the first place whether chromatin interactions actually are altered as a result of MED12 dysregulation. Similarly, AP-1 should be perturbed through AP-1 chemical inhibitors or shRNA to see whether CDK8/MED12 is disrupted and chromatin interactions.

2. How are differential chromatin interactions called in a statistically rigorous manner? This is a

major problem in the chromatin interaction field, that we cannot be sure whether a chromatin interaction is not present because of lack of depth of sequencing or because it does not exist. How do the authors address this challenge?

3. Hi-C is capable of detecting translocations and other genomic alterations. I expect Promoter Hi-C should be able to detect such alterations too, except these events would only be detected near promoters. Are any of these seen in the fibroids?

Reviewer #3 (Remarks to the Author):

Manuscript Review: NCOMMS-19-15990-T

In this manuscript, Moyo et al. have undertaken to examine differences in the epigenetic status and higher-order chromatin conformational state as a potential basis for gene expression changes between MED12-mutation positive uterine leiomyomas (LEIO) and corresponding normal myometrium (MYO). To this end, the authors performed integrative analyses based on RNA-seq as well as H3K27ac-specific ChIP-seq and promoter capture Hi-C from patient matched LEIO and MM tissues. This analysis revealed disease-associated changes in chromatin acetylation and enhancer (E)-promoter (P) interaction strength, epigenetic alterations proposed by the authors to underlie transcriptional dysregulation, including altered expression of ECM gene expression programs, in MED12-mutant LEIO. Subsequent genome-wide analysis of AP-1 chromatin association in both MYO and LEIO revealed predominant reduced E association of AP-1 (consistent with its downregulation as revealed by RNA-seq) in LEIO that were positively correlated with reduced acetylation as well as gain or loss of E-P contacts. Finally, genome-wide chromatin binding profiling of CDK8 submodule subunits MED12 and CDK8 revealed an association with Es in a manner that positively correlated with both acetylation and AP-1 occupancy, with no evidence for global loss of CDK8 submodule chromatin binding affinity that would otherwise indicate a MED12 mutation-driven binding defect. Taken together, the authors conclude that loss of AP-1 driven E-mediated control of ECM expression programs is an important mechanism in LEIO disease pathogenesis.

This is a comprehensive, compelling, and rigorously executed study that is certainly of interest to those in the gynecological health and disease research communities. The authors are to be commended for what is an exceptionally thorough and meticulous analysis of both epigenetic and higher-order chromatin architectural changes in MYO and LEIO. As the first such analysis of its kind as it relates to LEIO, this study therefore provides a new window into chromatin features that distinguish MYO from LEIO. Nonetheless, in its current iteration, several issues limit the broader impact of this study and its likely appeal to a wider readership.

First, the extreme focus on epigenetic profiling, while certainly laudable and informative, nonetheless results in a study that is largely descriptive – one based almost entirely on correlative relationships between altered epigenetic features and gene expression changes between MYO and LEIO. Presently, there is extremely limited experimental data presented to validate, much less establish causal relationships, between epigenetic alterations and gene expression changes, rendering the study lacking in mechanistic insights into disease etiology. For example, while Es are indeed characterized by specific epigenetic marks, including H3K27ac, they are, in the end, functional units. Yet, there is little data presented to validate functional connections between AP-1 regulated Es and their target genes in MYO cells and no efforts to establish whether AP-1 loss triggers altered acetylation and/or E-P contacts leading to altered expression of ECM genes as predicted by computational analyses. In the absence of such data, the study becomes simply a compendium of epigenetic alterations in normal and disease tissues from which mechanistic insight

may only be inferred. Certainly, this is a notable achievement and an important advance in its own right, but one that nonetheless fall short of providing real mechanistic insight into disease pathogenesis that might appeal to a more general audience.

A second concern relates to the fact that the sheer volume of epigenetic data presented is dense and often difficult to interpret from a biological perspective as it potentially relates to disease etiology. As written, the reader is deluged with a flood of large-scale computationally derived epigenomic data, much of it guilt by association without proven causation, that often obscures biologically relevant information. As one example, the conclusion that modified enhancer architecture, characterized by changes in E acetylation, E-P interaction strength, and differential E usage, drives aberrant transcription at differentially expressed genes in LEIO is a central theme that the authors wish to drive home in this study. However, the manuscript sections describing this data is packed with an overwhelming amount of genomic data referring to total numbers of genome-wide E-P and P-P contacts, along with those that overlap unique promoter distal regions that are acetylated and differentially acetylated. The sheer numbers and descriptions often read as simply a catalog of genomic alterations as opposed to a cogent description of those alterations that are likely to be biologically meaningful and important for differential gene expression. In other instances, it is not clear whether observed changes in E architecture refer specifically to genes differentially expressed between MYO and LEIO, or the genome more broadly, rendering conclusions regarding their relative contribution to pathologic gene dysregulation difficult to interpret. Accordingly, some editing of the data descriptions and figures so as to reduce the sheer volume of data, along with greater emphasis on epigenetic changes linked more specifically to differentially expressed genes in MYO and LEIO, might help the average reader to distinguish the "wheat from the chaff".

Finally, there some additional concerns with the data presented that require clarification to justify the author's conclusions.

Specific aspects of the study that relate to the concerns raised above are discussed below:

1. Transcriptome profiling of MYO and LEIO

(i) Although comparative gene expression profiling derived from RNA-seq-based analyses of MYO and LEIO tissues offers little new insight concerning transcriptional programs dysregulated between the two tissues (see for ex. Mehine et al. 2016 PNAS 113: 1315-1320), it is understood that these data are crucial for integrative analysis along with ChIP-seq and promoter-capture Hi-C analyses from the same tissues. In this regard, an important question is whether differences in gene regulation (and, indeed, E-P acetylation and E-P contacts described in subsequent sections) observed between MYO and LEIO tissue are specific to the MED12 mutant LEIO subclass, or alternatively, whether these differences are more generally reflective of LEIO disease state irrespective of the specific underlying driver mutation. Dysregulation of ECM gene expression programs appears to be a feature common to LEIO generally. While it is not suggested that the authors need expand their analysis to include MED12 WT LEIO, some discussion of this point seems appropriate, particularly given prior comparative profiling efforts in MYO as well as both MED12 WT and mutant LEIO. How does the author's gene expression profiling data compare with prior studies, and can any inferences be drawn concerning dysregulation of gene expression programs that might be specific to the MED12 mutant subclass?

2. Differential E acetylation as a dominant driver of transcriptional dysregulation.

(i) What is the justification for a sample number of 5 for MYO and LEIO tissues for ChIP-seq and Hi-C analyses? Is this number sufficient to establish statistically significant differences? Is H3K27ac sufficient alone to distinguish enhancers from promoters? What about H3K4me? What is definition of "enhancer" and "promoter" as defined by H3K27ac in this study? Clarification of these issues is recommended.

(i) Data in Fig. 2c and Fig. 3b are used to justify in part the perceived predominant role of differential enhancer vs promoter acetylation as a driver of gene dysregulation in MYO and LEIO. What statistical considerations were applied to these data sets to justify the conclusion of the authors that differential enhancer acetylation is more important than differential promoter acetylation as a determinant of differential gene regulation?

(ii) In the same vein, data in Supplementary Fig. 3c is also used to justify the conclusion that differential E acetylation is a predominant determinant of differential gene regulation in MYO vs LEIO. In the text, the authors refer to 1,835 Es associated with induced or repressed genes exhibiting little to no changes in promoter acetylation. However, from the heatmap in Supplementary Fig S3c (showing 3,183 differentially expressed genes with altered levels of E and P acetylation), it appears that the differentially expressed gene class exhibiting altered levels of E acetylation with no change in P acetylation is a quantitatively minor one (and within this class, the relationship between E acetylation and gene expression is not always positively correlated). Indeed, from this figure, the major class of differentially expressed genes appears to comprise those that carry alterations in both E and P acetylation. How do the authors reconcile this data with their conclusion that E, and not P, acetylation change is the dominant driver of differential gene regulation in MYO vs LEIO?

3. Altered E-P contact strength as a determinant of transcriptional dysregulation.

(i) The authors state that 25% of altered E-P contacts were associated with differentially expressed genes (Fig. 3c). Subsequent analysis of altered contacts overlapping Es with differential acetylation revealed several classes of modified cis-regulatory promoter distal regions (Fig. 3e). It is not clear from the description whether this analysis involved only Es specifically associated with differentially expressed genes, or alternatively, all Es across the genome. This distinction is a very important one to assess the role of altered E-P contacts in regulating differential gene expression in MYO and LEIO. In this regard, among differentially expressed genes, what was the proportional distribution of the different classes observed?

(ii) 3 different classes of modified cis-regulatory promoter distal regions were identified based on acetylation status and E-P contact strength (Supplementary Fig. S3d) : (1) those with unaltered E-P contact and differential acetylation; (2) those with altered E-P contact and unchanged acetylation, and (3) those with altered E-P contact and differential acetylation. What does this mean? What is the significance of this observation? Without some broader context, it is simply a catalog of chromatin alterations with little biological insight into disease etiology. In fact, the major class of modified cis-regulatory promoter-distal regions are those [class (1)] that correspond to regions with unaltered E-P contacts and differential H3K27ac. The minor class [(2)] of modified cis-regulatory promoter-distal regions are those that correspond to regions with altered E-P contacts and differential H3K27ac. So what is to be made of the relationship between altered enhancer-promoter contacts, acetylation, and gene expression, particularly for genes differentially expressed in MYO and LEIO?

4. Differential AP-1 enhancer binding and gene dysregulation in normal and disease tissue.

(i) The authors indicate that 761 E regions with differentially bound FOS or JUN were identified as directly interacting with 420 differentially expressed gene promoters, which include ECM genes and previously identified AP-1 target genes (Fig. 4f). Are there any correlations between E and P acetylation changes at these specific genes? In this regard, while the authors show a correlation between acetylation change and differential FOS/JUN binding among what appears to be all FOS/JUN binding sites, what about the specific E binding sites linked to differentially expressed gene promoters? And what of possible acetylation changes at these promoters? In other words, is differential FOS/JUN binding correlated with E, but not promoter, acetylation on AP-1 target genes differentially expressed in MYO and LEIO, as the authors seem to conclude?

(ii) The authors also state that: "AP-1-depleted E sites that coincided with altered E-P contacts were also identified, with an increase in JUN and FOS binding affinity being associated with an increase in E-P contact strength (Supplementary Fig. S6a, b). In contrast, a decrease in binding affinity was associated with both a gain and a loss in E-P contact strength." How does this data relate specifically to the 761 E regions with differentially bound FOS or JUN sites that directly interact with the promoters of 420 differentially expressed genes described previously? Are not these E-P contacts those that are most likely responsible for altered gene regulation in MYO vs LEIO? Describing relationships between E-P contacts and altered AP-1 binding on a genome-wide scale while ignoring these relationships at the specific subset of AP-1-linked Es most likely responsible for differential gene dysregulation clouds the central issue and buries the reader in yet more large-scale data sets that are difficult to interpret in a biologically and/or mechanistically meaningful way for disease etiology.

(iii) The authors provide examples of two differentially expressed genes, ADAM19 and EFEMP1, that show changes in enhancer binding of JUN and FOS (Fig. 4h,i). These genes showed a correlation between AP-1 binding and gene expression. How common is this phenomenon? What percentage of diff expressed ECM or AP-1 target genes showed this correlation? Is it statistically significant? All in all, there is a significant amount of data based on an exceptional effort level to be mined from these experiments focused on AP-1. However, from the perspective of this reviewer, the data are largely presented in manner that precludes meaningful biological or mechanistic interpretations for the disease. In other words, AP-1 binding is up and down and correlates with both increases and decreases in E-P contacts on a genome-wide scale. But how does this relate to genes specifically dysregulyated in MYO vs LEIO? Are their apparent meaningful trends or links to genes specifically dysregulated in MYO vs LEIO that might provide some measure of patho/biolgoical or mechanistic insight into the disease?

5. CDK8 subcomplex chromatin binding

(i) The observation that changes in CDK8 submodule chromatin association in LEIO is not consistent with a global loss of recruitment as predicted by a MED12 mutation-dependent interaction defect with CyclinC/CDK8 is an important finding that helps to clarify the basis by MED12 mutations drive LEIO. Furthermore, the finding that changes in CDK8 submodule chromatin association are positively correlated with changes in chromatin association of JUN and FOS is very interesting. Is the model here that AP-1 recruits Mediator containing the CDK8 submodule to AP-1 target genes? The authors could easily test this in MYO cells following AP-1 knockdown. In fact, many of the computationally derived observations in this study could be validated in the AP-1 knockdown model in MYO cells. This seems like a missed opportunity to validate and explore important functional connections between AP-1 regulated Es and their associated target genes. For example, would AP-1 depletion alter E acetylation and/or E-P contacts at AP-1 regulated ECM genes in a manner consistent with their altered expression? Furthermore, is Mediator recruitment by AP-1 required for E acetylation and/or the establishment of E-P contacts that drive AP-1 target gene transcription?

Reviewers' comments:

Reviewer #1 (Remarks to the Author):

Moyo and colleagues studied MED12 mutant uterine leiomyomas from 15 patients to initially identify differentially expressed genes in leiomyomas as compared to adjacent non-diseased myometrium. Indeed, they found an astonishing number of close to 6000 differentially regulated genes, which is about one quarter of all genes that are encoded in the human genome. Then, they performed ChIP-sequencing for the H3K27ac mark to find differentially acetylated regions in the genome. Again, they find a substantial fraction of 30% differentially acetylated regions in the genome. Using promoter capture Hi-C they find that the enhancer architecture is greatly altered in MED12-mutant uterine leiomyomas and is associated with a depletion of AP-1 occupancy on chromatin. They then establish primary myometrium cells and silence several subunits of the AP-1 complex. Indeed, gene sets are dysregulated in the knockdown conditions similar to the situation in leiomyomas. The authors conclude that AP-1 would be the driver of aberrant enhancer regulation and that this would be an important mechanism of leiomyoma pathogenesis.

Comment: It does not become fully clear from the study what the drivers of the epigenetic changes and altered enhancer-promoter interactions are that go together with reduced AP-1 occupancy. Are these changes indeed the drivers or rather passenger events in the disease?

Response: Mechanistic studies to determine drivers in uterine leiomyoma pathogenesis are still in their infancy. In fact, with the exception of the *MED12* mutation, which has been demonstrated to be a driver/ causal mutation by the use of a *MED12* mutant mouse model, it is unclear if other key observations such as loss of AP-1 gene expression, genetic aberrations affecting *HMG2*, *FH* and type IV collagens, are necessary for the development of the disease. Our manuscript attempts to begin to explain how AP-1 may be involved in leiomyoma disease development through alterations in enhancer function.

New experiments involving CRISPR/Cas9 mediated depletion of AP-1 subunits in human uterine smooth muscle cells (**Figure 6a-f, and supplementary figure 8a**) demonstrates that loss of AP-1 itself results in large-scale changes in enhancer acetylation as well as dysregulation of a large number of genes in uterine muscle cells and in that sense drives/ plays an important role in altering the epigenetic and transcriptomic state in leiomyoma disease pathogenesis. This work is a significant and necessary first step that adds rationale for further exploration of AP-1's role in an animal model. However, with the new experiments, we believe that this study has demonstrated two central themes in this paper:

1. We have shown that loss of AP-1 is sufficient to cause the large-scale changes seen in enhancer acetylation in leiomyoma tissue samples (**Figure 6d,e**).
2. We have demonstrated that loss of AP-1 is sufficient to cause the large-scale dysregulation of genes, including ECM genes (**Figure 6a-c and supplementary figure 8b-d**). Given that aberrant ECM deposition is a major phenotype of uterine leiomyomas, we believe that in this sense AP-1 is indeed a driver in this disease. In addition, the dysregulation of nearly 1,900 genes upon AP-1 depletion shows that AP-1 regulates a significant portion of the transcriptome and may explain the large number of dysregulated genes seen in leiomyoma.

Comment: The authors specifically analyzed leiomyomas carrying a particular mutation in

the MED12 gene. This mutation has previously been described as a gain-of-function mutation which was sufficient to induce leiomyomas and genomic instability in mice (PMID:26193636). This work and that recently published on lincRNA H19 (PMID:31089260) should be regarded by the authors and the respective findings at least be discussed in the context of their own observations. In the latter paper, TET3 activity on DNA is described in the context of leiomyomas, which might contribute to the interpretation of some findings that are reported here. For example, could there be a relation between de-/methylation of DNA and AP1 activity in the context of leiomyoma (even if the AP1 motif does not contain a CpG dinucleotide)? What is the relation between de-/methylation of DNA and the binding of the Mediator complex and the regulation of PolII transcription?

Response: Thank you for the comment. We have explored the expression of *H19* and *TET* family of genes in our uterine tissue samples. We observe increased *H19* expression in leiomyoma tissue samples and observe increased *H19* expression upon silencing of AP-1 in both primary cells cultured from myometrium tissue samples we obtained from hysterectomies performed at Prentice Women's Hospital as well as human uterine smooth muscle cells (HUtSMC) obtained from American Type Culture Collection (ATCC). This places AP-1 above *H19* in the mechanistic pathway of leiomyoma disease pathogenesis and further emphasizes the important role of AP-1 in this disease. Discussion on this topic is now included in the discussion of the revised manuscript (**Page 21, paragraph 3**).

Comment: On page 15 the authors state that “the observation of alterations in enhancer-promoter contacts and the identification of differential enhancer usage in uterine leiomyomas establishes the remodeling of enhancer regions not only important in normal enhancer-mediated regulation of gene expression but also key in the development of disease.” The evidence that is provided here does not really allow to conclude whether these alterations are causal or rather mediator/symptom for the disease.

Response: The reviewer is correct. We revised the sentence as follows “Taken together, the observations of alterations in H3K27Ac signal at distal sites, altered promoter contacts, and the identification of differential enhancer usage in uterine leiomyomas, all suggest that in addition to important promoter-proximal gene regulatory events, enhancer malfunction, a previously unexplored but significant area in leiomyoma disease pathogenesis, is an important mechanism of gene dysregulation in uterine leiomyomas.” (**Page 19, paragraph 1**)

Comment: The link between the activating mutation in MED12 via H19 and TET3 could help to better explain some of the findings that are described in the current manuscript.

Response: The reviewer mentions the recent study that shows regulation of *MED12* gene expression by *H19*. However, we would like to point out that changes in *MED12* expression have not been shown in uterine fibroids. Only mutation of *MED12* has been observed and this does not result in up-regulation or down-regulation of *MED12*. Also, neither H19 nor TET3 were demonstrated to cause genetic lesions in *MED12* in the study pointed out by the reviewer. In addition, we would also like to point out that despite a tremendous amount of focus on DNA methylation in uterine leiomyomas, only 120 dysregulated genes were identified with hypo/hypermethylated DNA in leiomyoma¹. Whilst we do not object to H19 and TET3 possibly being involved in altered DNA methylation patterns in fibroids, we do not believe that there is any evidence to date that *H19*, *TET3*, or DNA methylation play a role in the development of mutations in *MED12*, which is the observed causal event in leiomyoma pathogenesis.

However, thanks to the reviewer's earlier comments, we have noted changes in H19, TET1 and TET3 in our leiomyoma tissue samples relative to myometrium and also noted an increase in *H19* expression upon AP-1 depletion and have included these observations in the manuscript (**Page 21, paragraph 3**).

Comment: Similarly, the authors claim (p16) that their “study describes the role of AP-1 in extracellular matrix gene regulation and the mechanisms by which decreased AP-1 expression leads to the formation of uterine leiomyomas.” The authors should tone down this statement in the light of the literature described above. The interesting question should be what leads to downregulation of AP-1 in leiomyomas. The authors should discuss others' findings in the context of their own.

Response: We have modified this statement and other statements in the manuscript to better reflect our results and their integration with relevant previously observed results. The statement now reads “This study describes the role of AP-1 in uterine muscle cell gene regulation and the mechanisms by which decreased AP-1 expression may contribute to the formation of uterine leiomyomas.” (**Page 19, paragraph 2**)

Comment: On page 12 the authors write that they wanted to assess whether exon2 mutations at G44 in MED12 protein would result in a loss of CDK8 submodule chromatin binding in uterine leiomyomas. This hot-spot gain-of-function mutation is the only recurrent mutation in the MED12 gene and has been observed in several malignancies. The mutation is indeed likely to alter the activity of the Mediator complex, however, this is unlikely to result in a loss of CDK8 submodule chromatin binding. Indeed, the authors observed different occupancy profiles of wildtype vs. mutant MED12, however, not a global loss of submodule chromatin binding. On the same page, they speak of a tissue type-specific CDK8 submodule occupancy in myometrium and leiomyoma (Supplementary Fig. 7e). The differences in hierarchical clustering might well be caused by different binding patterns of wt vs. mutant MED12. If true, this would be similar to some mutations (in the DNA binding-domain) in TP53 which have been shown to alter the spectrum of binding sites the respective mutant forms of p53 have compared to wt p53.

Response: Multiple mutations in *MED12* have been identified, in exon 2 as well as other loci in leiomyomas as well as in other diseases²⁻⁵. With respect to the mutation in G44, unlike in p53, the mutation exists in a region where cyclin C – CDK8 would bind. Therefore, if there was an effect caused by the mutation occurring in the cyclin C – CDK8 binding pocket, the mutation should affect the binding of the submodule uniformly and thus result in either an increased CCNC-CDK8 interaction with MED12 or a decreased interaction with MED12, the latter of which has been demonstrated *in vitro*. However, in tissues, this is not what we observe. Instead, we observe changes in CDK8 submodule that better mirror the changes in enhancer acetylation, suggesting that the submodule, in its role as a Mediator transcriptional co-factor complex, is simply tracking the changes in the transcriptional complex.

Comment: On page 17, the authors write that “MED13 is a stabilizing protein in the submodule that compensates for any MED12 mutant dependent physical interaction defect but fails to compensate for the defect in kinase activity” (citing references 61 and 62). The role mutant MED12 has in the formation of leiomyomas speaks against a fully efficient compensation of the altered activity by MED13. A mutant dependent interaction defect would typically be observed in a tumor suppressor gene. The hot-spot nature of the mutation

and the published effects mutant MED12 induces rather speak for an oncogenic function the protein acquires by the mutation. The authors should rephrase their statement.

Response: The statement “MED13 is a stabilizing protein in the submodule that compensates for any MED12 mutant dependent physical interaction defect but fails to compensate for the defect in kinase activity” is a paraphrased quote of the previously referenced studies, in which the stabilizing effect of MED13 was experimentally determined^{6,7}. Park et al. state in their abstract that “Furthermore, we find that within the Mediator kinase module, MED13 directly binds to the MED12 C terminus, thereby suppressing an apparent UF mutation–induced conformational change in MED12 that otherwise disrupts its association with CycC-CDK8/19. Thus, in the presence of MED13, mutant MED12 can bind, but cannot activate, CycC-CDK8/19.”⁶ We believe that the binding profile of CDK8 and MED12 revealed in our work supports this interpretation.

The point we are trying to make is not that MED13 compensates for loss in CDK8 submodule kinase activity but rather that MED13 sufficiently stabilizes the CDK8-CCNC-MED12-MED13 complex in the event of a mutation in MED12, that in the absence of MED13 would cause a loss of binding of CCNC-CDK8 to MED12. What the referenced studies show however, is that although MED13 stabilizes the complex, it does not compensate for the loss of kinase activity in the mutant state. Hence, although the mutation does not cause a mutation-dependent change in the CDK8 submodule chromatin binding profile, it may and probably does cause a loss of kinase activity and as such further work to determine the kinase substrate will be important as well as how this substrate is involved in the disease pathogenesis.

The reviewer also states “A mutant dependent interaction defect would typically be observed in a tumor suppressor gene.” What this work suggests is that this mutation-dependent defect is not seen in this case, even if in other cases it would typically be observed in a tumor suppressor gene. MED12 may acquire an oncogenic function through the mutation, but this work suggests it is not through a CDK8 submodule-binding defect. Perhaps the mutation causes a binding defect in another partner protein in leiomyomas but we refrain from making such a suggestion, as it would be highly speculative at this stage.

Comment: In a follow-up study it might be interesting to investigate also patients who are wildtype for MED12 in the respective leiomyoma. This might help to even better decipher the direct mechanisms that are driven by mutant MED12 and to potentially uncover alternative mechanisms of leiomyoma development. Is loss of AP1 also prevalent in wt MED12 leiomyomas? Is the chromatin stable there?

The same hot-spot mutation in codon 12 has been found also in 1/3 of soft tissue tumors, 12% of breast cancers, and ~5% of skin cancers and other tumor entities. Overexpression of MED12 has been detected in tumors of the urinary tract, stomach cancer, soft tissue tumors, oesophageal cancer, and tumors of the large intestine, among others (Cosmic database). It will be interesting to see whether similar changes in the epigenetic landscape of these tumors exist and if similar mechanisms are relevant also in other tumor diseases.

Response: The reviewer is correct and indeed, this is part of the ongoing work in the lab. We are currently working on epigenomic profiling of leiomyoma tumors that overexpress HMGA2 or have mutations in *FH*. It is noted that previous data sets show down regulation of AP-1 in leiomyomas of other subtypes. We will look to investigate changes in AP-1 binding

on chromatin with the hope of determining if the features described in this work are specific to MED12 mutant tumors or apply to leiomyomas in general. Additionally, glycine 44 of *MED12* is also frequently mutated in breast fibroadenomas⁸. Similar to uterine leiomyomas, breast fibroadenomas are also benign, hormone dependent, and the most common tumor in this tissue type. It would be interesting to see if AP-1 is down regulated in these tumors.

Minor

Comment: On page 10 the authors write that Fig. 4c should show expression of JUN, FOS and ATF families of genes. The data is shown for JUN and FOS, however, not for members of the ATF family of transcription factors.

Response: Gene tracks for ATF3 and ATF6 have been added as requested (**Supplementary figure 5a**).

Comment: On page 20 the authors should provide a g-force when they speak of low speed centrifugation. Below they should add a temperature when they describe brief sonication of samples using a Misonix device.

Response: Methods have been amended to reflect speed of centrifugation during cell lysis for tissue ChIP experiments as well as sonication in an ice water bath (**Page 25**).

Comment: On page 23 they describe the viral transduction of concentrated lentiviral particles for silencing AP-1 gene or non-silencing control. The MOI should be provided.

Response: The MOI was not determined for this experiment, however the methods have been expanded to explain how viral amount was determined to be appropriate for the transduction (**Page 28**).

Comment: Which software was used to align ChIP and Chi-C reads to the genome (page 24)? The next paragraph speaks of STAR aligner for RNA-seq data.

Response: The software used to align ChIP and Chi-C reads to the genome are HOMER (v4.10.3) and (HiCUP, v0.5.10). Information regarding data-specific analysis such as read alignment is reported under the relevant experiment's data analysis. In this case, information regarding ChIP and Chi-C read alignments are reported under the respective headings in the methods: "ChIP-seq data processing and analysis" and "Chi-C data processing and analysis." (**Pages 33 and 34**)

Reviewer #2 (Remarks to the Author):

In this manuscript, Moyo et al. investigate epigenetic alterations (including chromatin interactions and transcription factor binding) at dysregulated transcripts in MED12 mutant fibroids. The concept of investigating chromatin interactions, which are known to be regulated by MED12, in fibroids, which are known to have dysregulated MED12, is novel and interesting. This paper is one of the first set of papers to investigate chromatin interactions in clinical samples, in this case, fibroids, and will be of interest in the field. The figures and text are well done.

Response: We thank the reviewer for recognizing the importance and significance of our work.

I have a few major issues to discuss:

Comment: 1. This paper is largely descriptive, yet the authors argue that the observations seen are a reflection of a MED12 dysregulation pattern. However, these observations are just correlations. To be able to say that the observations of altered chromatin interactions result from MED12 dysregulation, the authors would need to perform MED12 alteration (e.g. CRISPR to remove exon 2) in a cell line or two and show altered chromatin interactions as a result.

Response: We agree that MED12 mutation, as demonstrated by the mouse model, is a driver event in the disease. However, whether this mutation is itself directly responsible for altered contacts and enhancer histone acetylation is undetermined and may not be fully addressed by an *in vitro* system as it would not determine if altered contacts and enhancers were direct effects of mutation in MED12 as opposed to an important but indirect, downstream effect such as a loss of AP-1. To complement our genome-wide strongly correlative studies, we have shown in new experiments (**Figure 6**) with AP-1 depleted cells that the effect of the MED12 mutation on enhancer regulation is most likely indirect, and mediated at least in part by AP-1. However, how mutation of MED12 leads to loss of AP-1 remains to be determined.

Comment: The authors mention in the discussion that they are unsure whether MED13 may compensate, and that the MED12 pathogenesis may be due to CDK8 kinase activity, however, it would still be useful to establish in the first place whether chromatin interactions actually are altered as a result of MED12 dysregulation. Similarly, AP-1 should be perturbed through AP-1 chemical inhibitors or shRNA to see whether CDK8/MED12 is disrupted and chromatin interactions.

Response: In new experiments, we have demonstrated through AP-1 depletion in human uterine smooth muscle cells that loss of AP-1 is important in altered enhancer epigenomic profile. AP-1 loss recapitulates changes in enhancer H3K27 acetylation seen in leiomyoma tissue samples compared to myometrium (**Figure 6c,d**).

Comment: 2. How are differential chromatin interactions called in a statistically rigorous manner? This is a major problem in the chromatin interaction field, that we cannot be sure whether a chromatin interaction is not present because of lack of depth of sequencing or because it does not exist. How do the authors address this challenge?

Response: The reviewer raises a good point. This was a major consideration when we opted to perform promoter capture Hi-C instead of conventional Hi-C for this study as previous studies demonstrate a 15-17 fold enrichment of promoter interactions in promoter capture Hi-C over conventional Hi-C. This helps avoid the problem of false negatives, whereby promoter contacts are not identified due to a lack of depth. We have also used 5 biological replicates, which also increases the odds of detecting valid chromatin interactions.

In addition, interaction confidence scores for promoter interacting regions were calculated using the Capture Hi-C Analysis of Genomic Organisation (CHiCAGO) pipeline⁹. A stringent contact score ≥ 5 has been demonstrated to be quite robust in identifying significant interactions, with a previous study of reciprocal promoter capture Hi-C of 949 promoter interacting regions demonstrating a score ≥ 5 performs very well at identifying high confidence interactions whilst minimizing the rate of false positives¹⁰.

With regard to differential chromatin interactions, only interactions with an average of at least 10 counts in either condition were used for differential contact calling. This is highlighted in the methods under the header “CHi-C data processing and analysis” on page 34.

Comment: 3. Hi-C is capable of detecting translocations and other genomic alterations. I expect Promoter Hi-C should be able to detect such alterations too, except these events would only be detected near promoters. Are any of these seen in the fibroids?

Response: Genomic alterations are already well documented in the field using whole genome sequencing at a 40X depth and chromosomal aberrations have been described in this disease

11

Reviewer #3 (Remarks to the Author):

Manuscript Review: NCOMMS-19-15990-T

In this manuscript, Moyo et al. have undertaken to examine differences in the epigenetic status and higher-order chromatin conformational state as a potential basis for gene expression changes between MED12-mutation positive uterine leiomyomas (LEIO) and corresponding normal myometrium (MYO). To this end, the authors performed integrative analyses based on RNA-seq as well as H3K27ac-specific ChIP-seq and promoter capture Hi-C from patient matched LEIO and MM tissues. This analysis revealed disease-associated changes in chromatin acetylation and enhancer (E)-promoter (P) interaction strength, epigenetic alterations proposed by the authors to underlie transcriptional dysregulation, including altered expression of ECM gene expression programs, in MED12-mutant LEIO. Subsequent genome-wide analysis of AP-1 chromatin association in both MYO and LEIO revealed predominant reduced E association of AP-1 (consistent with its downregulation as revealed by RNA-seq) in LEIO that were positively correlated with reduced acetylation as well as gain or loss of E-P contacts. Finally, genome-wide chromatin binding profiling of CDK8 submodule subunits MED12 and CDK8 revealed an association with Es in a manner that positively correlated with both acetylation and AP-1 occupancy, with no evidence for global loss of CDK8 submodule chromatin binding affinity that would otherwise indicate a MED12 mutation-driven binding defect. Taken together, the authors conclude that loss of AP-1 driven E-mediated control of ECM expression programs is an important mechanism in LEIO disease pathogenesis.

This is a comprehensive, compelling, and rigorously executed study that is certainly of interest to those in the gynecological health and disease research communities. The authors are to be commended for what is an exceptionally thorough and meticulous analysis of both epigenetic and higher-order chromatin architectural changes in MYO and LEIO. As the first such analysis of its kind as it relates to LEIO, this study therefore provides a new window into chromatin features that distinguish MYO from LEIO. Nonetheless, in its current iteration, several issues limit the broader impact of this study and its likely appeal to a wider readership.

Response: Thank you for finding our work comprehensive and compelling. We do hope and believe that the chromatin features in myometrium and leiomyoma identified in this work will be useful to the field. Below are our responses to your comments.

Comment: First, the extreme focus on epigenetic profiling, while certainly laudable and informative, nonetheless results in a study that is largely descriptive – one based almost entirely on correlative relationships between altered epigenetic features and gene expression changes between MYO and LEIO. Presently, there is extremely limited experimental data presented to validate, much less establish causal relationships, between epigenetic alterations and gene expression changes, rendering the study lacking in mechanistic insights into disease etiology. For example, while Es are indeed characterized by specific epigenetic marks, including H3K27ac, they are, in the end, functional units. Yet, there is little data presented to validate functional connections between AP-1 regulated Es and their target genes in MYO cells and no efforts to establish whether AP-1 loss triggers altered acetylation and/or E-P contacts leading to altered expression of ECM genes as predicted by computational analyses. In the absence of such data, the study becomes simply a compendium of epigenetic alterations in normal and disease tissues from which mechanistic insight may only be inferred. Certainly, this is a notable achievement and an important advance in its own right, but one that nonetheless fall short of providing real mechanistic insight into disease pathogenesis that might appeal to a more general audience.

Response: This is well noted and we have addressed this by performing additional RNA-seq and H3K27Ac ChIP-seq on AP-1 depleted cells. This new work is described in **figure 6** and is especially compelling given the number of dysregulated genes shown by the RNA-seq. Importantly, the loss of AP-1 subunit expression resulted in large-scale changes in the H3K27Ac cistrome of AP-1 depleted cells, primarily occurring at distal intergenic regions. Moreover, these changes mirrored those seen in leiomyoma tissue samples and thereby implicating AP-1 loss in uterine muscle cell enhancer dysfunction. We think this exciting new data, as requested by the reviewers, establishes causality between AP-1 expression and enhancer state in uterine muscle cells, with data showing conclusively that loss of AP-1 results in an altered enhancer cistrome, the profile of which is strikingly similar to that seen in uterine fibroids.

Comment: A second concern relates to the fact that the sheer volume of epigenetic data presented is dense and often difficult to interpret from a biological perspective as it potentially relates to disease etiology. As written, the reader is deluged with a flood of large-scale computationally derived epigenomic data, much of it guilt by association without proven causation, that often obscures biologically relevant information. As one example, the conclusion that modified enhancer architecture, characterized by changes in E acetylation, E-P interaction strength, and differential E usage, drives aberrant transcription at differentially expressed genes in LEIO is a central theme that the authors wish to drive home in this study. However, the manuscript sections describing this data is packed with an overwhelming amount of genomic data referring to total numbers of genome-wide E-P and P-P contacts, along with those that overlap unique promoter distal regions that are acetylated and differentially acetylated. The sheer numbers and descriptions often read as simply a catalog of genomic alterations as opposed to a cogent description of those alterations that are likely to be biologically meaningful and important for differential gene expression. In other instances, it is not clear whether observed changes in E architecture refer specifically to genes differentially expressed between MYO and LEIO, or the genome more broadly, rendering conclusions regarding their relative contribution to pathologic gene dysregulation difficult to interpret. Accordingly, some editing of the data descriptions and figures so as to reduce the sheer volume of data, along with greater emphasis on epigenetic changes linked more

specifically to differentially expressed genes in MYO and LEIO, might help the average reader to distinguish the "wheat from the chaff".

Response: We understand these concerns and we welcome the reviewer's comments on this, as it was a point of significant internal discussion before our initial submission: How much to present and where to focus? However, we would like to point out that this conundrum is common to all integrative genome-wide studies. In addition, given that this is the first promoter capture Hi-C data in uterine fibroids, and more broadly in tissue samples, our intent was to provide a broad and thorough description of the data set and the results so as to allow the reader to be able to determine the quality of the data without having to re-analyze the data themselves. This also goes towards our intent that this study be used not only as a first of its kind study in fibroids that will spur further research into the role of enhancer regulation and AP-1 in leiomyomas, but also as a rich data resource for the field in general. This study involved the generation of a lot of data that we feel would be under utilized if not fully explored by us as well as others. However, in agreement with the reviewer's request, we have made modifications to each section to better highlight the important biological features and reduce some of the details, especially surrounding the promoter capture Hi-C. It must be noted though that even with these modifications, it may simply be impossible to avoid the density of information thrown at the reader as this is an attempt to integrate 3 orthogonal data sets that each generated a lot of information.

Finally, although there is a significant overlap as well as strong correlation between RNA-seq, ChIP-seq and promoter capture Hi-C results, there are also many features that are not concordant. The size of these makes it unlikely that they are artifacts. Although how they directly relate to dysregulated gene expression is as yet unclear and whilst we might not fully understand the significance of these epigenomic events, we feel it is important to discuss these features in order to bring them to the attention of other researchers.

Finally, there some additional concerns with the data presented that require clarification to justify the author's conclusions.

Specific aspects of the study that relate to the concerns raised above are discussed below:

1. Transcriptome profiling of MYO and LEIO

Comment: (i) Although comparative gene expression profiling derived from RNA-seq-based analyses of MYO and LEIO tissues offers little new insight concerning transcriptional programs dysregulated between the two tissues (see for ex. Mehine et al. 2016 PNAS 113: 1315-1320), it is understood that these data are crucial for integrative analysis along with ChIP-seq and promoter-capture Hi-C analyses from the same tissues. In this regard, an important question is whether differences in gene regulation (and, indeed, E-P acetylation and E-P contacts described in subsequent sections) observed between MYO and LEIO tissue are specific to the MED12 mutant LEIO subclass, or alternatively, whether these differences are more generally reflective of LEIO disease state irrespective of the specific underlying driver mutation. Dysregulation of ECM gene expression programs appears to be a feature common to LEIO generally. While it is not suggested that the authors need expand their analysis to include MED12 WT LEIO, some discussion of this point seems appropriate, particularly given prior comparative profiling efforts in MYO as well as both MED12 WT and mutant LEIO. How does the author's gene expression profiling data compare with prior

studies, and can any inferences be drawn concerning dysregulation of gene expression programs that might be specific to the MED12 mutant subclass?

Response: We agree with the reviewer and are currently working on epigenomic profiling of leiomyoma tumors that overexpress HMGA2 or have mutations in fumarate hydratase. It is noted that previous data sets show down regulation of AP-1 in leiomyomas of other subtypes. We will look to investigate changes in AP-1 binding on chromatin in non-MED12 mutant leiomyomas with the hope of determining if the features described in this work are specific to MED12 mutant tumors or apply to leiomyomas in general. We have included discussion on this point in the discussion (**Page 20, paragraph 3**).

In addition, we have compared our RNA-seq data set to Mehine et al. as requested. We find that at a q -value of 0.05, our data set and MED12-subtype data from the Mehine et al. data set recover 8,285 and 7,953 differentially expressed genes respectively. However, only a 56% overlap of differentially expressed genes is seen between the two data sets. Additionally, Mehine et al. reports 258 genes with > 2 fold change in expression, compared to 3,389 differentially expressed genes with > 2 fold change (FDR < 0.05) in our data set, with 225/258 of the Mehine et al. set contained in our set, which also include JUN, JUNB, FOS, and FOSB. Differences in technique (i.e. microarray vs. RNA-seq) and unavailability of the raw data preclude further analysis of the Mehine et al. data set.

2. Differential E acetylation as a dominant driver of transcriptional dysregulation.

Comment: (i) What is the justification for a sample number of 5 for MYO and LEIO tissues for ChIP-seq and Hi-C analyses? Is this number sufficient to establish statistically significant differences?

Response: Histone modification ChIP-seq and capture Hi-C necessitate a trade-off between the number of replicates and the sequencing depth. While 5 biological replicates might not have sufficient power to make conclusions about the population in general, we believe that an n of 5 provides us with a sufficient number of biological replicates to be able to state that the phenomena we are investigating is real and reproducible and at the same time be able to achieve a high enough sequencing depth to identify true ChIP peaks and Hi-C contacts.

Comment: Is H3K27ac sufficient alone to distinguish enhancers from promoters? What about H3K4me? What is definition of "enhancer" and "promoter" as defined by H3K27ac in this study? Clarification of these issues is recommended.

Response: Promoters are defined as regions from -1000bp to +100bp of the transcriptional start site and enhancers regions are H3K27Ac peaks that fall outside promoter regions and are further subdivided into H3K27Ac peaks found in exonic, intronic, and intergenic regions. We have also clarified this definition in the methods (**Page 33**). H3K27Ac has been shown to be a very robust marker of enhancers and promoters. Given evidence in previous studies of strong overlap between H3K4me3 and H3K27Ac cistromes around promoters (-1000bp to +100bp), we believe that H3K27Ac is sufficient to identify active promoters, especially when viewed in conjunction with RNA-seq data showing actively transcribed regions at known transcriptional start sites. Although H3K4me1 has been shown to be a marker of enhancer regions, recent studies have demonstrated that this histone mark is dispensable for enhancer function¹².

Comment: (i) Data in Fig. 2c and Fig. 3b are used to justify in part the perceived predominant role of differential enhancer vs promoter acetylation as a driver of gene dysregulation in MYO and LEIO. What statistical considerations were applied to these data sets to justify the conclusion of the authors that differential enhancer acetylation is more important than differential promoter acetylation as a determinant of differential gene regulation?

Response: Fig. 2d and Fig. 3b in the revised manuscript are heatscatter plots meant to provide graphical representation of individual H3K27Ac regions that are described in Fig. 2c in the revised manuscript i.e. pie chart showing differentially acetylated genomic loci. What Fig. 2c reflects is that altered acetylation signal is occurring at different genomic loci, including gene promoters, but is disproportionately occurring at intergenic sites. This difference has been statistically determined to be significant (chi-squared). We believe this study demonstrates that although promoter acetylation, which has previously been demonstrated to be important for gene transcription, is observed in myometrium and leiomyoma, a significant proportion (46%) of differentially expressed genes show little to no change in promoter acetylation signal. The changes in enhancer acetylation seen at both enhancers associated with genes with differential promoter acetylation as well as enhancers associated with genes with unchanged promoter acetylation, explains some of the gene dysregulation that may seem inexplicable given unaltered promoter acetylation.

Note: The order of Figures 2c and 2d in the original manuscript has now been reversed in the revised manuscript, as we believe this order provides better flow of the results.

Comment: (ii) In the same vein, data in Supplementary Fig. 3c is also used to justify the conclusion that differential E acetylation is a predominant determinant of differential gene regulation in MYO vs LEIO. In the text, the authors refer to 1,835 Es associated with induced or repressed genes exhibiting little to no changes in promoter acetylation. However, from the heatmap in Supplementary Fig S3c (showing 3,183 differentially expressed genes with altered levels of E and P acetylation), it appears that the differentially expressed gene class exhibiting altered levels of E acetylation with no change in P acetylation is a quantitatively minor one (and within this class, the relationship between E acetylation and gene expression is not always positively correlated). Indeed, from this figure, the major class of differentially expressed genes appears to comprise those that carry alterations in both E and P acetylation. How do the authors reconcile this data with their conclusion that E, and not P, acetylation change is the dominant driver of differential gene regulation in MYO vs LEIO?

Response: We do not suggest that enhancer acetylation is more important than events occurring at promoters nor do we suggest that enhancer acetylation is the dominant driver of gene regulation. Rather, given that many diseases are rarely caused by any one factor, we simply intend to highlight a previously unexplored but significant area in leiomyoma disease pathogenesis i.e. enhancer malfunction. We certainly do believe changes in promoter profiles of transcribed genes are very important as evidenced by changes in promoter-proximal H3K27 acetylation in leiomyomas at many genes, however, we believe that this new data shows a significant role that is played by enhancer malfunction in the disease. We have sought to clarify this in various sections of the manuscript (**Page 8, paragraph 1; Page 9, paragraph 1; Page 19, paragraph 1**)

3. Altered E-P contact strength as a determinant of transcriptional dysregulation.

Comment: (i) The authors state that 25% of altered E-P contacts were associated with

differentially expressed genes (Fig. 3c). Subsequent analysis of altered contacts overlapping Es with differential acetylation revealed several classes of modified cis-regulatory promoter distal regions (Fig. 3e). It is not clear from the description whether this analysis involved only Es specifically associated with differentially expressed genes, or alternatively, all Es across the genome. This distinction is a very important one to assess the role of altered E-P contacts in regulating differential gene expression in MYO and LEIO. In this regard, among differentially expressed genes, what was the proportional distribution of the different classes observed?

Response: We have clarified the number of altered enhancers that have differential H3K27Ac that are associated with dysregulated genes. This section now reads “A survey of all altered contacts overlapping enhancers with differential H3K27Ac signal shows that regions with increased acetylation are associated with increased promoter contact strength whereas regions depleted of H3K27Ac signal are linked to both contacts with increasing and decreasing strength (Fig. 3e). Approximately 42% (521/1,229) of all altered contacts overlapping enhancers with differential H3K27Ac signal are associated with differentially expressed genes.” (Page 9, paragraph 2)

Comment: (ii) 3 different classes of modified cis-regulatory promoter distal regions were identified based on acetylation status and E-P contact strength (Supplementary Fig. S3d) : (1) those with unaltered E-P contact and differential acetylation; (2) those with altered E-P contact and unchanged acetylation, and (3) those with altered E-P contact and differential acetylation. What does this mean? What is the significance of this observation? Without some broader context, it is simply a catalog of chromatin alterations with little biological insight into disease etiology. In fact, the major class of modified cis-regulatory promoter-distal regions are those [class (1)] that correspond to regions with unaltered E-P contacts and differential H3K27ac. The minor class [(2)] of modified cis-regulatory promoter-distal regions are those that correspond to regions with altered E-P contacts and differential H3K27ac. So what is to be made of the relationship between altered enhancer-promoter contacts, acetylation, and gene expression, particularly for genes differentially expressed in MYO and LEIO?

Response: As previous studies have demonstrated, most enhancer-promoter contacts are cell lineage defining and as such are stable regardless of gene expression changes. However, despite the majority of contacts being stable, a subset are dynamic and do correlate with major events in the cell i.e. changes in gene expression of important factors. However, although demonstrated to be important events in cellular processes such as cell differentiation, this group of enhancers is still in the minority. The novelty of this finding is not the size of the group but its previously defined role. Hence, given the fact that leiomyomas, although distinct from normal myometrium, are still smooth muscle cells, it is the expected result that regions with unaltered enhancer-promoter contact strength and differential H3K27 acetylation overwhelmingly constitute the largest group of modified enhancers. We have clarified this further in the results section (Page 10, paragraph 1).

In summary, it is not the goal of this work to append the pre-existing conclusions that the majority of chromatin loops are stable. Rather, it is our intention to highlight that the small but significant group of dynamic contacts are not only observed in biological processes such as cell differentiation but are also seen in disease states and we believe that this is part of the novelty of this work; dynamic contacts are observed in a disease state and similar to

differentiation, may be important and play a role in disease pathogenesis.

4. Differential AP-1 enhancer binding and gene dysregulation in normal and disease tissue.

Comment: (i) The authors indicate that 761 E regions with differentially bound FOS or JUN were identified as directly interacting with 420 differentially expressed gene promoters, which include ECM genes and previously identified AP-1 target genes (Fig. 4f). Are there any correlations between E and P acetylation changes at these specific genes?

Response: We replaced the scatter plot (Fig. 4f, original manuscript) showing the relationship between AP-1 enhancer occupancy and gene expression with two heat maps in the revised manuscript, for FOS (Fig. 4g) and JUN (Fig. 4h) showing the relationship between these enhancer AP-1 and H3K27Ac signal with gene expression changes at enhancers contacting promoters of differential expressed genes.

Comment: In this regard, while the authors show a correlation between acetylation change and differential FOS/JUN binding among what appears to be all FOS/JUN binding sites, what about the specific E binding sites linked to differentially expressed gene promoters? And what of possible acetylation changes at these promoters? In other words, is differential FOS/JUN binding correlated with E, but not promoter, acetylation on AP-1 target genes differentially expressed in MYO and LEIO, as the authors seem to conclude?

Response: As stated in response to point 2ii above, we do not suggest that changes in enhancer architecture are solely associated with dysregulated genes that have unchanged promoter acetylation. As a matter of fact, dysregulated genes with changing promoter acetylation as well as dysregulated genes with unchanged promoter acetylation are both associated with altered enhancer architecture. We are simply highlighting that enhancer architecture, a previously unexplored area of dysregulation in uterine leiomyomas, is significantly altered in the disease and in the case of dysregulated genes with unchanged promoter acetylation, altered enhancer architecture may explain the changes in gene expression. We show that in altered enhancer architecture, AP-1 is a major enriched factor for both genes with changing and unchanged promoter acetylation. In conclusion, the new data demonstrates that loss of AP-1 is sufficient to partially recapitulate the changes in enhancer acetylation seen in leiomyomas.

Comment: (ii) The authors also state that: “AP-1-depleted E sites that coincided with altered E-P contacts were also identified, with an increase in JUN and FOS binding affinity being associated with an increase in E-P contact strength (Supplementary Fig. S6a, b). In contrast, a decrease in binding affinity was associated with both a gain and a loss in E-P contact strength.” How does this data relate specifically to the 761 E regions with differentially bound FOS or JUN sites that directly interact with the promoters of 420 differentially expressed genes described previously? Are not these E-P contacts those that are most likely responsible for altered gene regulation in MYO vs LEIO? Describing relationships between E-P contacts and altered AP-1 binding on a genome-wide scale while ignoring these relationships at the specific subset of AP-1-linked Es most likely responsible for differential gene dysregulation clouds the central issue and buries the reader in yet more large-scale data sets that are difficult to interpret in a biologically and/or mechanistically meaningful way for disease etiology.

(iii) The authors provide examples of two differentially expressed genes, ADAM19 and

EFEMP1, that show changes in enhancer binding of JUN and FOS (Fig. 4h,i). These genes showed a correlation between AP-1 binding and gene expression. How common is this phenomenon? What percentage of diff expressed ECM or AP-1 target genes showed this correlation? Is it statistically significant? All in all, there is a significant amount of data based on an exceptional effort level to be mined from these experiments focused on AP-1. However, from the perspective of this reviewer, the data are largely presented in manner that precludes meaningful biological or mechanistic interpretations for the disease. In other words, AP-1 binding is up and down and correlates with both increases and decreases in E-P contacts on a genome-wide scale. But how does this relate to genes specifically dysregulated in MYO vs LEIO? Are their apparent meaningful trends or links to genes specifically dysregulated in MYO vs LEIO that might provide some measure of patho/biological or mechanistic insight into the disease?

Response to ii and iii: We would like to point out that not only a positive correlation, as shown by examples in **figures 4i** and **j**, are the biologically meaningful events to be noted, however, as shown in **figures 4g** and **4h**, the majority depleted AP-1 binding sites are associated with both a loss as well as an increase in gene expression. The purpose of meticulously cataloguing changes in the epigenome is meant to highlight the fact that you cannot necessarily group the effects of AP-1 loss into either a loss or a gain in enhancer acetylation, and more importantly, a loss or a gain in gene expression. We also believe that the new data in AP-1 depleted cells further emphasizes this point. AP-1 loss causes both an increase well as a decrease in enhancer acetylation at enhancers. As such, the focus on meaningful trends is misplaced; the overarching theme is that large-scale enhancer dysregulation/ reorganization is correlated with loss of AP-1 and this loss alone, as evidenced by H3K27Ac ChIP-seq in AP-1 depleted cells, is sufficient to explain a lot of these large-scale changes. That is the hypothesis we set out to test and we believe the data substantiates our claims.

We believe the genome-wide scale data is the appropriate level to look at the impact of AP-1 in this disease and we believe the results in AP-1 depleted cells substantiate this approach. With regard to ECM associated genes, we believe that as aberrant ECM deposition is a major feature of the disease, it is an important feature to highlight in the dataset. However, as mentioned in the discussion (**Page 17, paragraph 2**), although a third of all known ECM-associated genes are dysregulated in leiomyomas, this only accounts for approximately 5.5% of all differentially expressed genes identified in the leiomyoma transcriptome. As such to focus solely on this group and the small number of differentially expressed genes with promoter-proximal AP-1 motifs, is to negate the importance of the rest of the altered transcriptome. We acknowledge that we did not make this clearer in the manuscript and we have modified it to better reflect what we believe should be the focus or the overarching story we are looking to tell i.e. Large-scale transcriptomic and epigenomic changes are observed in uterine leiomyomas and loss of AP-1 is a significant causal event in these changes. A subset of these changes affects direct ECM target genes.

5. CDK8 subcomplex chromatin binding

Comment: (i) The observation that changes in CDK8 submodule chromatin association in LEIO is not consistent with a global loss of recruitment as predicted by a MED12 mutation-dependent interaction defect with CyclinC/CDK8 is an important finding that helps to clarify the basis by MED12 mutations drive LEIO. Furthermore, the finding that changes in CDK8 submodule chromatin association are positively correlated with changes in chromatin

association of JUN and FOS is very interesting. Is the model here that AP-1 recruits Mediator containing the CDK8 submodule to AP-1 target genes? The authors could easily test this in MYO cells following AP-1 knockdown.

Response: We believe that loss of AP-1 subunit expression is downstream of MED12/CDK8 submodule in the mechanistic pathway. We concur with the generally accepted hypothesis that exon 2 mutations in *MED12* drive leiomyoma pathogenesis and although this is unlikely to be due to a submodule-binding defect based on this work and previous studies, the mutation likely causes loss of CDK8 submodule kinase activity. We hypothesize that loss of AP-1 expression is either directly caused by loss of CDK8 kinase activity or indirectly through dysregulation of an AP-1 regulator that is impacted by loss of CDK8 kinase activity. However, we believe to test this requires a whole new line of experimentation specifically geared towards determination of the effects of a kinase deficient submodule involving the determination of the kinase substrate of CDK8 in uterine muscle cells. We are currently pursuing such lines of experimentation but feel this would fall outside the scope of this study.

Comment: In fact, many of the computationally derived observations in this study could be validated in the AP-1 knockdown model in MYO cells. This seems like a missed opportunity to validate and explore important functional connections between AP-1 regulated Es and their associated target genes. For example, would AP-1 depletion alter E acetylation and/or E-P contacts at AP-1 regulated ECM genes in a manner consistent with their altered expression? Furthermore, is Mediator recruitment by AP-1 required for E acetylation and/or the establishment of E-P contacts that drive AP-1 target gene transcription?

Response: As per reviewer's suggestion we have performed the experiment (**Figure 6a-f**). We believe that the new experiments in AP-1 depleted cells addresses dependency of E acetylation on AP-1.

In summary we hope that the reviewers will agree that we have sincerely addressed all of their major questions/suggestions by including additional experiments and altering texts as necessary. We also hope that all reviewers will find the revised manuscript acceptable for publication in Nature Communications.

References

1. Maekawa, R. *et al.* Genome-wide DNA methylation analysis reveals a potential mechanism for the pathogenesis and development of uterine leiomyomas. *PLoS ONE* **8**, e66632 (2013).
2. Rump, P. *et al.* A novel mutation in MED12 causes FG syndrome (Opitz-Kaveggia syndrome). *Clin. Genet.* **79**, 183–188 (2011).
3. Vulto-van Silfhout, A. T. *et al.* Mutations in MED12 cause X-linked Ohdo syndrome. *Am. J. Hum. Genet.* **92**, 401–406 (2013).
4. Rishg, H. *et al.* A recurrent mutation in MED12 leading to R961W causes Opitz-Kaveggia syndrome. *Nat. Genet.* **39**, 451–453 (2007).
5. Barbieri, C. E. *et al.* Exome sequencing identifies recurrent SPOP, FOXA1 and MED12 mutations in prostate cancer. *Nat. Genet.* **44**, 685–689 (2012).
6. Park, M. J. *et al.* Oncogenic exon 2 mutations in Mediator subunit MED12 disrupt allosteric activation of cyclin C-CDK8/19. *J. Biol. Chem.* **293**, 4870–4882 (2018).
7. Turunen, M. *et al.* Uterine leiomyoma-linked MED12 mutations disrupt mediator-associated CDK activity. *CellReports* **7**, 654–660 (2014).

8. Lim, W. K. *et al.* Exome sequencing identifies highly recurrent MED12 somatic mutations in breast fibroadenoma. *Nat. Genet.* **46**, 877–880 (2014).
9. Cairns, J. *et al.* CHiCAGO: robust detection of DNA looping interactions in Capture Hi-C data. *Genome Biol.* **17**, 127 (2016).
10. Javierre, B. M. *et al.* Lineage-Specific Genome Architecture Links Enhancers and Non-coding Disease Variants to Target Gene Promoters. *Cell* **167**, 1369–1384.e19 (2016).
11. Mehine, M. *et al.* Characterization of Uterine Leiomyomas by Whole-Genome Sequencing. *N Engl J Med* **369**, 43–53 (2013).
12. Dorigi, K. M. *et al.* Mll3 and Mll4 Facilitate Enhancer RNA Synthesis and Transcription from Promoters Independently of H3K4 Monomethylation. *Molecular Cell* **66**, 568–576.e4 (2017).

REVIEWERS' COMMENTS:

Reviewer #1 (Remarks to the Author):

The authors have sufficiently responded to the vast majority of the comments. They have improved their study also by knocking out five members of the AP-1 complex in uterine smooth muscle cells and finding that these perturbations induce similar alterations in H3K27Ac as well as gene expression as have been observed in patient samples. Still, the finding that some enhancer/promoter elements are positively while others are negatively regulated by AP-1 persists, leaving the notion that the study is mostly descriptive. Further, supplementary Figure 8a suggests that knockout efficiency was not high for most JUN-members while this was better for FOS. Similarly, shRNA-mediated knockdown of FOS seems to be better than that of JUN (Supplementary Figure 8b). The authors should discuss their observations in the light of the different knockout/knockdown efficiencies. The data that is presented in supplementary figure 8 hence does not fully support a "loss" of AP-1 that could explain the changes in enhancer acetylation seen in leiomyoma tissue samples. Loss of JUN-protein might not be tolerated by the cells. Higher efficiency of FOS knockout/knockdown might shift the binding partner of JUN to some other AP-1 member protein (ATF?), making their experimental system somewhat difficult to interpret.

I still struggle with the statement on p5: „This suggests that the changes in gene expression between normal and MED12 mutant disease tissue types are primarily attributable to biological pathways that are important for the development and maintenance of the leiomyoma disease state.“ Changes in gene expression programs do indeed correlate with disease state. The difference between correlation and causality has been a matter of debate since the development of gene expression arrays >20 years ago. I regard it as unlikely that all those 5,831 differentially expressed genes are indeed causally involved in leiomyoma disease etiology.

Reviewer #2 (Remarks to the Author):

All my comments have been addressed.

Reviewer #3 (Remarks to the Author):

In their revised manuscript, Moyo et al. have responded satisfactorily to this reviewer's initial concerns. The revised manuscript is an improved version, and represents a thorough and meticulous analysis of epigenetic and higher-order chromatin architectural changes occurring in normal versus uterine fibroid disease tissue. As the first such analysis of its kind as it relates to uterine fibroids, this study therefore provides a new window into chromatin features that distinguish normal from the diseased state, and thus fills an important gap between the known genetic and unknown epigenetic changes that contribute to uterine fibroid pathogenesis. I therefore support publication of the manuscript in its revised form, contingent upon clarification of two points:

1. CDK8 submodule binding: The authors state in both the results (page 13) and discussion (page 22) sections that prior published studies suggest that mutations in MED12 exon 2 may result in a decrease in CDK8 submodule binding affinity at sites of active transcription. Moreover, in the discussion section (page 22), the authors state that "studies have suggested that MED12 mutations may affect transcription either through the disruption of CDK8 kinase activity or through decreased binding affinity of the MED12-containing CDK8 submodule at sites of active transcription". In fact, this statement poses a false dichotomy. First, the initial report suggesting that mutations in MED12 disrupt its association with CycC-CDK8/19 would not be predicted to

decrease chromatin association of the entire CDK8 submodule, but only CycC-CDK8/19. This is due to the fact that MED13 links MED12 to core Mediator. Second, in any event, the initial report that mutations in MED12 disrupt its association with CycC-CDK8/19 were subsequently shown in more recently published studies to most likely result from experiments performed in vitro in the absence of MED13 or in cells overexpressing WT or mutant MED12 derivatives. In fact, these more recently published studies, based on analysis of Mediator composition and activity in patient-derived myometrium and uterine fibroid tissues, concluded that mutations in MED12 disrupt Mediator kinase activity without altering Mediator composition (i.e., disruption of the MED12 interaction with CycC-CDK8/19). Accordingly, the findings of the authors in the present study that CDK8 recruitment to chromatin "is not consistent with a global loss of recruitment that would result from a mutation-dependent MED12-Cyclin C-CDK8 interaction defect" supports prior published studies and the text should be edited to reflect this fact.

2. CRISPR/Cas9 experiments: The description of the CRISPR/Cas9 experiments could be clarified to render it clearer. More specifically, upon initial reading, it was not clear what CRISPR-targeted cells were specifically used for analysis (i.e. individual gene-targeted cells or a pool of individually gene-targeted cells). It was only clear upon reading the methods section that cells combinatorially targeted for AP-1 subunits were used. Perhaps the addition of a single word to the description on page 15 would be preferable: "...To test this, JUN and FOS family of genes were combinatorially (or simultaneously) silenced in human uterine smooth muscle cells..."

Thomas Boyer

REVIEWERS' COMMENTS:

Reviewer #1 (Remarks to the Author):

Comment: The authors have sufficiently responded to the vast majority of the comments. They have improved their study also by knocking out five members of the AP-1 complex in uterine smooth muscle cells and finding that these perturbations induce similar alterations in H3K27Ac as well as gene expression as have been observed in patient samples. Still, the finding that some enhancer/promoter elements are positively while others are negatively regulated by AP-1 persists, leaving the notion that the study is mostly descriptive. Further, supplementary Figure 8a suggests that knockout efficiency was not high for most JUN-members while this was better for FOS. Similarly, shRNA-mediated knockdown of FOS seems to be better than that of JUN (Supplementary Figure 8b). The authors should discuss their observations in the light of the different knockout/knockdown efficiencies. The data that is presented in supplementary figure 8 hence does not fully support a “loss” of AP-1 that could explain the changes in enhancer acetylation seen in leiomyoma tissue samples. Loss of JUN-protein might not be tolerated by the cells. Higher efficiency of FOS knockout/knock might shift the binding partner of JUN to some other AP-1 member protein (ATF?), making their experimental system somewhat difficult to interpret.

Response: The reviewer’s point regarding AP-1 depletion efficiencies is well noted. However, higher efficiency of FOS depletion resulting in a shift in JUN binding partners is only one possible outcome. The lower efficiency in JUN family silencing may explain why we see less differentially acetylated regions in HUtSMCs as compared to leiomyoma tissue samples and a more robust depletion may result in a greater recapitulation of enhancer malfunction as compared to tissue samples. We have now included a discussion on both possibilities on page 19, paragraph 2.

Comment: I still struggle with the statement on p5: “This suggests that the changes in gene expression between normal and MED12 mutant disease tissue types are primarily attributable to biological pathways that are important for the development and maintenance of the leiomyoma disease state.” Changes in gene expression programs do indeed correlate with disease state. The difference between correlation and causality has been a matter of debate since the development of gene expression arrays >20 years ago. I regard it as unlikely that all those 5,831 differentially expressed genes are indeed causally involved in leiomyoma disease etiology.

Response: We agree with the reviewer that it is possible, and perhaps even likely, that some of the differentially expressed genes identified in leiomyoma tissue samples are secondary effects and others are perhaps not causal. We have amended the discussion on page 16, paragraph 1, with the statement “It will be key going forward to delineate expression changes in genes that are important in leiomyoma pathogenesis from possible non-causal secondary effects.”

However, we still believe that the statement on page 5 the reviewer refers to is still valid. While there may still be room for philosophical debate regarding correlation vs. causality as it pertains to differential gene expression, we believe that the many studies over the past two decades do in fact support a hypothesis that changes in gene expression that correlate with disease state do

generally prove to be causal. The question going forward is which of these gene expression changes are causal.

Reviewer #2 (Remarks to the Author):

All my comments have been addressed.

Reviewer #3 (Remarks to the Author):

In their revised manuscript, Moyo et al. have responded satisfactorily to this reviewer's initial concerns. The revised manuscript is an improved version, and represents a thorough and meticulous analysis of epigenetic and higher-order chromatin architectural changes occurring in normal versus uterine fibroid disease tissue. As the first such analysis of its kind as it relates to uterine fibroids, this study therefore provides a new window into chromatin features that distinguish normal from the diseased state, and thus fills an important gap between the known genetic and unknown epigenetic changes that contribute to uterine fibroid pathogenesis. I therefore support publication of the manuscript in its revised form, contingent upon clarification of two points:

Comment: CDK8 submodule binding: The authors state in both the results (page 13) and discussion (page 22) sections that prior published studies suggest that mutations in MED12 exon 2 may result in a decrease in CDK8 submodule binding affinity at sites of active transcription. Moreover, in the discussion section (page 22), the authors state that "studies have suggested that MED12 mutations may affect transcription either through the disruption of CDK8 kinase activity or through decreased binding affinity of the MED12-containing CDK8 submodule at sites of active transcription". In fact, this statement poses a false dichotomy. First, the initial report suggesting that mutations in MED12 disrupt its association with CycC-CDK8/19 would not be predicted to decrease chromatin association of the entire CDK8 submodule, but only CycC-CDK8/19. This is due to the fact that MED13 links MED12 to core Mediator.

Response: We agree with the reviewer that MED12 exon 2 mutations would not result in decreased chromatin association of the whole complex. We are aware that the mutation in MED12 exists in a region where cyclin C would bind and as such would result in a disruption of cyclin C-CDK8 interaction with MED12/MED13 and the rest of mediator. Stating that MED12 mutation would result in a loss of CDK8 submodule binding was perhaps an inaccurate generalization which we have now corrected by specifically referencing the loss in cyclin C-CDK8 binding, the main catalytic subunits of the CDK8 submodule, instead of describing it as the loss in CDK8 submodule chromatin occupancy.

Comment: Second, in any event, the initial report that mutations in MED12 disrupt its association with CycC-CDK8/19 were subsequently shown in more recently published studies to most likely result from experiments performed in vitro in the absence of MED13 or in cells overexpressing WT or mutant MED12 derivatives. In fact, these more recently published studies, based on analysis of Mediator composition and activity in patient-derived myometrium and uterine fibroid tissues, concluded that mutations in MED12 disrupt Mediator kinase activity without altering Mediator composition (i.e., disruption of the MED12 interaction with CycC-

CDK8/19). Accordingly, the findings of the authors in the present study that CDK8 recruitment to chromatin “is not consistent with a global loss of recruitment that would result from a mutation-dependent MED12-Cyclin C-CDK8 interaction defect” supports prior published studies and the text should be edited to reflect this fact.

Response: We believe the text already reflects the reviewer’s views and as such does not need to be modified. The later published work referred to by the reviewer is referenced in the same paragraph on page 20 that the reviewer refers to.

Both the earlier work by Turunen et al. and the more recent work by Park et al. are referenced and we believe the text in its current form clearly states that our data supports the conclusions in Park et al. which the reviewer refers to. In short, the point made by the reviewer is already stated in the page 20, paragraph 1.

Comment: CRISPR/Cas9 experiments: The description of the CRISPR/Cas9 experiments could be clarified to render it clearer. More specifically, upon initial reading, it was not clear what CRISPR-targeted cells were specifically used for analysis (i.e. individual gene-targeted cells or a pool of individually gene-targeted cells). It was only clear upon reading the methods section that cells combinatorially targeted for AP-1 subunits were used. Perhaps the addition of a single word to the description on page 15 would be preferable: “...To test this, JUN and FOS family of genes were combinatorially (or simultaneously) silenced in human uterine smooth muscle cells...”

Response: The text has been modified to read “To test this, *JUN* and *FOS* family of genes were simultaneously silenced in primary human uterine smooth muscle cells”